# Conditional internalization of PEGylated nanomedicines by PEG engagers for triple negative breast cancer therapy

Yu-Cheng Su[1], Pierre-Alain Burnouf[1,2], Kuo-Hsiang Chuang[3], Bing-Mae Chen[1], Tian-Lu Cheng[4] & Steve R. Roffler[1,5]

Triple-negative breast cancer (TNBC) lacks effective treatment options due to the absence of traditional therapeutic targets. The epidermal growth factor receptor (EGFR) has emerged as a promising target for TNBC therapy because it is overexpressed in about 50% of TNBC patients. Here we describe a PEG engager that simultaneously binds polyethylene glycol and EGFR to deliver PEGylated nanomedicines to EGFR$^+$ TNBC. The PEG engager displays conditional internalization by remaining on the surface of TNBC cells until contact with PEGylated nanocarriers triggers rapid engulfment of nanocargos. PEG engager enhances the anti-proliferative activity of PEG-liposomal doxorubicin to EGFR$^+$ TNBC cells by up to 100-fold with potency dependent on EGFR expression levels. The PEG engager significantly increases retention of fluorescent PEG probes and enhances the antitumour activity of PEGylated liposomal doxorubicin in human TNBC xenografts. PEG engagers with specificity for EGFR are promising for improved treatment of EGFR$^+$ TNBC patients.

[1] Institute of Biomedical Sciences, Academia Sinica, Taipei 11529, Taiwan. [2] Taiwan International Graduate Program in Molecular Medicine, National Yang-Ming University and Academia Sinica, Taipei 11529, Taiwan. [3] Graduate Institute of Pharmacognosy, Taipei Medical University, Taipei 11031, Taiwan. [4] Department of Biomedical Science and Environmental Biology, Center for Biomarkers and Biotech Drugs, Kaohsiung Medical University, Kaohsiung 80708, Taiwan. [5] Graduate Institute of Medicine, Kaohsiung Medical University, Kaohsiung 80708, Taiwan. Correspondence and requests for materials should be addressed to T.-L.C. (email: tlcheng@kmu.edu.tw) or to S.R.R. (email: sroff@ibms.sinica.edu.tw).

Breast cancer is the second most common cancer in the world. Triple-negative breast cancer (TNBC), which comprises 11.2–16.3% of breast cancers, lacks expression of oestrogen receptors, progesterone receptors and human epidermal growth factor receptor 2. TNBC is typically heterogeneous, aggressive and is associated with poor prognosis with limited treatment options due to the absence of well-defined therapeutic targets[1,2]. Systemic chemotherapy has been the primary treatment option for TNBC until it was discovered that epidermal growth factor receptor (EGFR) is overexpressed in ~50% of TNBC tumours[3]. EGFR-targeted agents are therefore under development for the treatment of TNBC[4]. However, EGFR-targeted tyrosine kinase inhibitors such as gefitinib[5] and erlotinib[6] display minimal therapeutic efficacy in TNBC patients.

Nanomedicines, including PEGylated liposomal doxorubicin, are currently under investigation for the treatment of TNBC[7–9]. Nanocarriers are attractive because they can alter the pharmacokinetic profile of drugs, reduce off-target toxicity and improve the therapeutic index[10]. Tumour accumulation of nanomedicines relies on the enhanced permeability and retention effect in which the leaky blood vasculature combined with impaired lymphatic drainage can facilitate passive accumulation of nanosized particles in tumours[11]. Lung, breast and ovarian tumours display high enhanced permeability and retention effect-mediated accumulation of nanocarriers, making nanomedicines an attractive treatment alternative for TNBC[12].

Active targeting by functionalizing the surface of nanocarriers with ligands that bind to endocytic receptors on cancer cells can promote receptor-mediated endocytosis for increased cellular uptake of nanomedicines with concomitant improved antitumour activity[13–16]. However, many technical and regulatory hurdles must be overcome to produce new nanocarriers with reproducible and homogenous ligand densities and activities[17]. Attachment of targeting ligands can also compromise the stealth features of nanocarriers and hinder tumour uptake[10,18].

Pre-targeting strategies can help curtail these problems by separating the production of the targeting moiety and nanocarrier as well as allowing the administration of unmodified highly stealth nanocarriers[19]. Because nanomedicines are commonly grafted with poly (ethylene glycol) (PEG) to decrease uptake and clearance by the reticuloendothelial system[20–22], here we develop a general pre-targeting strategy for conditional internalization of PEGylated nanomedicines for improved treatment of EGFR[+] TNBC. This is accomplished by generating bispecific PEG-binding antibodies (PEG engagers) for targeted delivery of PEGylated nanomedicines to tumours. Pre-targeting of PEG engagers induce endocytosis of PEGylated nanocarriers into EGFR[+] TNBC cancers leading to enhanced antitumour efficacy of PEG-modified therapeutic agents *in vitro* and *in vivo*.

## Results

**Characterization of bispecific PEG engagers**. A bispecific PEG engager was generated by genetically fusing a humanized anti-PEG Fab fragment with an anti-EGFR single-chain disulfide-stabilized Fv fragment. The PEG engager was designed to bind to EGFR on TNBC cells but remain dormant until contact and binding to PEG-coated nanocarriers induces rapid internalization into cancer cells (Fig. 1). Briefly, a humanized anti-PEG (6.3) Fab was constructed as a single open reading frame by fusing variable light chain with constant kappa light chain ($V_L$–$C_\kappa$) and variable heavy chain with first constant heavy chain ($V_H$–$CH_1$) domains with an internal ribosome entry site bicistronic expression linker, allowing the coordinated expression of light and heavy chains. Single chain disulfide-stabilized variable fragments (dsFv) specific for CD19 (negative control) or EGFR were linked to the C terminus of the 6.3 Fab via a GGGGS peptide linker to generate bispecific PEG engager[CD19] and PEG engager[EGFR] (Fig. 2a). The PEG engager genes were inserted into a lentiviral expression vector and stable 293FT producer cells were generated[41]. PEG engagers purified from the culture medium display the expected molecular sizes as visualized on a reducing and a non-reducing 10% sodium dodecyl sulfate polyacrylamide gel electrophoresis (SDS–PAGE) (Fig. 2b). The PEG engager[CD19] and PEG engager[EGFR] have molecular weights of 78 kDa and 79 kDa, respectively, as determined by matrix-assisted laser desorption/ionization time-of-flight mass spectrometry (Fig. 2c). Size-exclusion high-performance liquid chromatography analysis showed a major peak corresponding to a monomer with minimal aggregation (Supplementary Fig. 1a). The melting temperatures of PEG engager[CD19] and PEG engager[EGFR] (75 °C and 75.8 °C, respectively), as determined by differential scanning calorimetry (Supplementary Fig. 1b), were higher than the typical melting temperatures of Fab fragments (61.9–69.4 °C)[23,24], indicating that the engagers display good thermal stabilities. Analysis of PEG engager binding to PEG by microscale thermophoresis showed that PEG engager[EGFR] (equilibrium dissociation constant, $K_D = 7.6 \pm 1.13$ nM) and PEG engager[CD19] ($K_D = 7.6 \pm 1.05$ nM) displayed similar binding affinities (Fig. 2d). The binding affinity ($K_D$) of the PEG engagers to their corresponding tumour antigen targets (EGFR and CD19) was $0.96 \pm 0.23$ nM and $3.6 \pm 0.35$ nM for PEG engager[EGFR] and PEG engager[CD19], respectively (Fig. 2e).

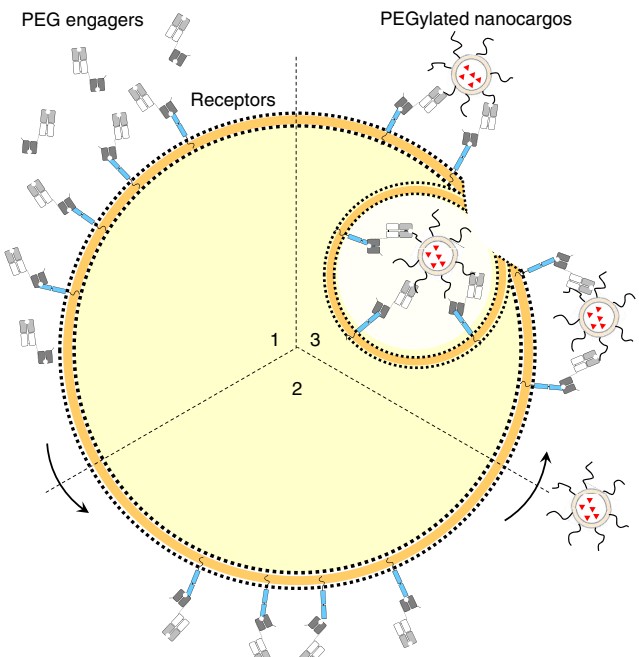

**Figure 1 | Overview of PEG engager pre-targeting strategy.** The pre-targeting approach relies on (1) PEG engagers (bispecific humanized anti-PEG Fab × antitumour antigen single chain disulfide-stabilized variable fragment) binding to target cancer receptors, (2) PEG engagers remaining on the cell surface until contact with (3) PEGylated nanoparticles trigger conditional internalization into the cells.

**Specific delivery of PEGylated nanoparticles by PEG engagers**. To determine whether PEG engager[EGFR] can specifically deliver PEGylated nanoparticles into EGFR-positive cancer cells, we investigated cancer cell lines with different expression levels of EGFR, including MCF7 EGFR-negative non-TNBC breast adenocarcinoma cells, MDA-MB-468 EGFR-positive TNBC cells

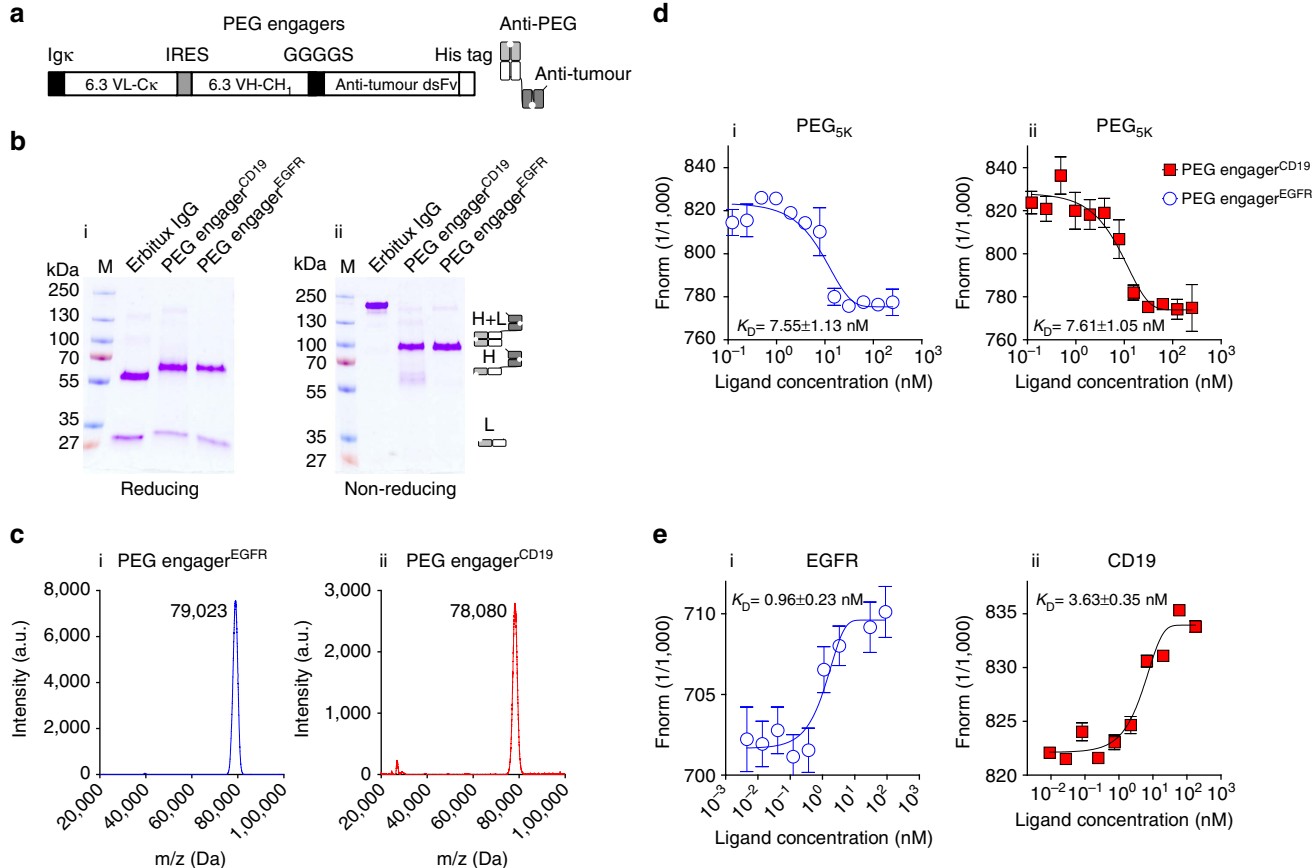

**Figure 2 | Production and analysis of recombinant PEG engagers.** (**a**) Schematic representation of PEG engager constructs, which code for a murine immunoglobulin kappa chain leader sequence (Igκ), a humanized 6.3 light chain (6.3 VL-Cκ), an internal ribosome entry site (IRES) sequence, a humanized 6.3 heavy chain fragment (6.3 VH–CH₁), a glycine-serine peptide linker (GGGGS), an antitumour single chain disulfide-stabilized variable fragments (dsFv, anti-CD19 or anti-EGFR), and a polyhistidine-tag (His tag). (**b**) Reducing (i) and non-reducing (ii) sodium dodecyl sulfate polyacrylamide gel electrophoresis showing Coomassie blue staining of Erbitux, PEG engager^EGFR and PEG engager^CD19. M, PageRuler prestained protein ladder (Fermentas). H (heavy chain), L (light chain). (**c**) Precise molecular weight of PEG engager^EGFR (i) and PEG engager^CD19 (ii) analysed by matrix-assisted laser desorption/ionization time-of-flight mass spectrometry. Mass-to-charge ratio (m/z). (**d**) Binding affinity of PEG engager^EGFR (i) and PEG engager^CD19 (ii) to PEG₅ₖ analysed by microscale thermophoresis (n = 3). (**e**) Binding affinity of PEG engager^EGFR to recombinant EGFR protein (i) and PEG engager^CD19 to recombinant CD19 protein (ii) analysed by microscale thermophoresis (n = 3). Equilibrium dissociation constant ($K_D$). Data are shown as mean ± s.d. Representative microscale thermophoresis data from three independent experiments are shown.

and A431 EGFR-positive epidermoid carcinoma cells. Cancer cell-specific uptake of PEGylated nanoparticles mediated by PEG engagers was examined by real-time confocal microscopy cell imaging of EGFR-negative MCF7 cells or EGFR-positive MDA-MB-468 and A431 cells treated stepwise with PEG engager^EGFR or PEG engager^CD19 and then fluorescent PEGylated Qtracker 655 non-targeted quantum dots (PEG-Qdot655). Both MDA-MB-468 TNBC and A431 cells express EGFR but not CD19. MCF7 cells express neither EGFR nor CD19. PEG engager^EGFR-mediated rapid accumulation of PEG-Qdot655 in both MDA-MB-468 and A431 cells (Fig. 3a,b(i–iii)), but not in MCF7 cells as shown in Fig. 3c(i–iii). By contrast, no uptake of PEG-Qdot655 was observed in MDA-MB-468, A431 and MCF7 cells treated with control PEG engager^CD19 as shown in Fig. 3(iv–vi). We conclude that PEG engager^EGFR can deliver PEGylated nanoparticles into cancer cells that express EGFR.

**Conditional internalization of PEGylated nanoparticles.** We examined whether PEG engager^EGFR could trigger conditional receptor-mediated internalization dependent on the presence of PEGylated nanoparticles. Alexa Fluor 647-labelled PEG engager^EGFR was first added to live cancer cells for 1 h. Confocal

imaging revealed that the engager remained on the plasma membrane of MDA-MB-468 (Fig. 4a(i–iii) and Supplementary Fig. 3ai) and BT-20 (Supplementary Fig. 2i–iv) cells at 37 °C for 1 h with almost no internalization (Alexa Fluor 647 appears in green pseudo color). PEG engager^EGFR displayed limited endocytosis in MDA-MB-486 cells even after 9 h (Supplementary Fig. 3bi). PEG-Qdot655 added to the cells, however, bound to PEG engager^EGFR on the cell membrane and were rapidly internalized into the cells (Fig. 4axii–xiv, Supplementary Fig. 2xii–xiv and Supplementary Fig. 3bvi–viii, red colour). Co-localization of PEG engager^EGFR and PEG-Qdot655 or lysosomes (Fig. 4a, Supplementary Figs 2 and 3b) verified that PEG engager^EGFR can conditionally stimulate endocytosis of PEGylated nanoparticles and then localize in lysosomes (Fig. 4b,c and Supplementary Fig. 3c).

**Efficacy of PEG engager-directed liposomal drugs *in vitro*.** We next investigated whether PEG engager^EGFR can enhance the anti-proliferation activity of a drug-loaded nanocarrier in different type of cancer cells that express either wild-type EGFR or mutated EGFR. MDA-MB-231, MDA-MB-468 and BT-20 are TNBC cancer cell lines that express wild-type EGFR. SKBR3 is a

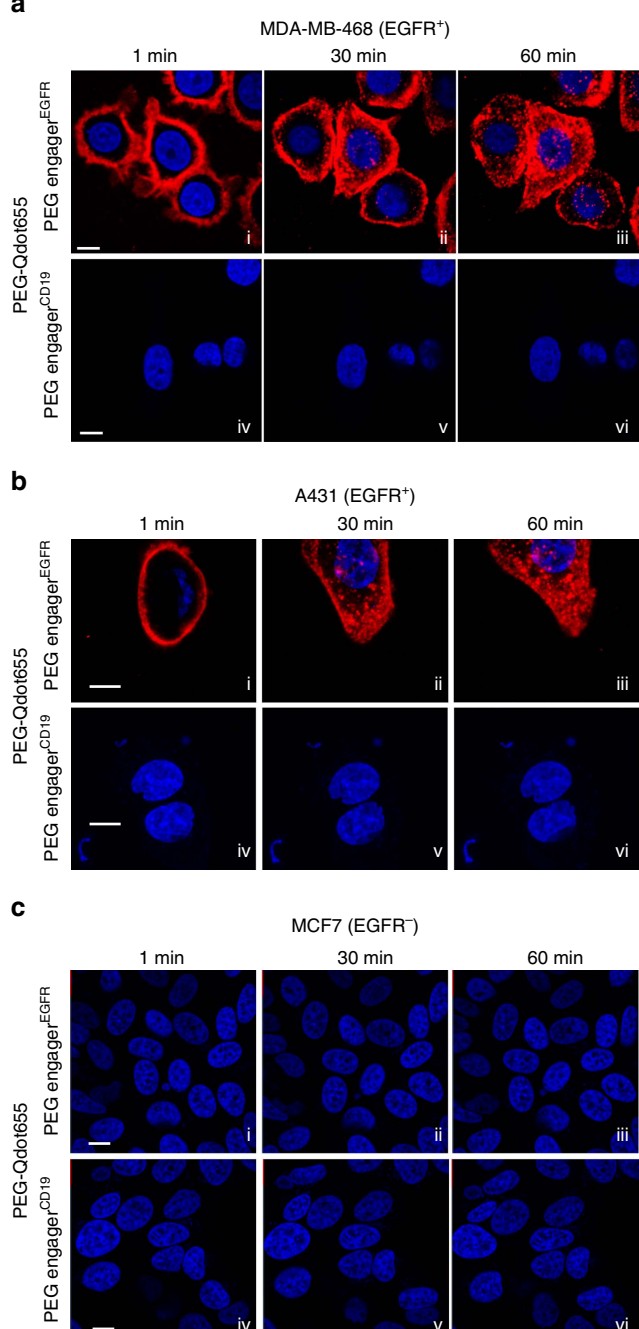

**Figure 3 | Dual antigen-binding activity of PEG engagers.** PEG engager[EGFR] (i–iii) and PEG engager[CD19] (iv–vi) supplemented with Hoechst 33342 (blue) were incubated with MDA-MB-468 (**a**), A431 (**b**) or MCF7 (**c**) cells followed by PEG-Qdot655 (red) and observed in real time with a digital confocal microscope. Scale bar, 10 µm. Representative confocal images from three independent experiments are shown.

non-TNBC breast adenocarcinoma cell line that expresses wild-type EGFR and PC9 is non-small cell lung cancer cell line that expresses EGFR with a delta E746-A750 deletion in the tyrosine kinase domain. These cells were incubated with PEG engager[EGFR] or PEG engager[CD19] as a negative control and subsequently treated with graded concentrations of free drug (doxorubicin or vinorelbine), empty liposomes, PEG-liposomal doxorubicin (Doxisome) or PEG-liposomal vinorelbine. PEG engager[EGFR] significantly enhanced the anti-proliferation activity of Doxisome

(Fig. 5 and Supplementary Fig. 4) and PEG-liposomal vinorelbine (Supplementary Fig. 5) against EGFR-positive cancer cells as compared with drug-loaded nanocarrier alone, drug-loaded nanocarrier plus PEG engager[CD19] or empty liposomes with PEG engager[EGFR]. The half maximal effective concentration (EC$_{50}$) values for PEG engager[EGFR] targeted Doxisome in BT-20, MDA-MB-468 and MDA-MB-231 cells were 101-fold, 74-fold and 107-fold lower than control PEG engager[CD19] targeted Doxisome, respectively (Fig. 5d). Neither PEG engager[EGFR] nor PEG engager[CD19] altered the sensitivity of HepG2 cells (EGFR negative) to Doxisome (Supplementary Fig. 4c). In contrast to wild-type BT-20 cells, PEG engager[EGFR] did not enhance the anti-proliferation activity of Doxisome in BT-20/shEGFR cancer cells (BT-20 cells treated with short hairpin RNA to knockdown the expression of EGFR) as compared with drug-loaded nano-carrier alone or drug-loaded nanocarrier plus PEG engager[CD19] (Supplementary Fig. 6), further showing that PEG engager[EGFR] mediates endocytosis of PEGylated nanocarriers via the EGFR internalization pathway. We conclude that PEG engager[EGFR] can increase the anticancer activity of PEGylated medicines (Doxisome and PEG-liposomal vinorelbine) to EGFR$^+$ cancer cells.

Pre-existing anti-PEG antibodies in healthy donors might negatively impact PEG engager targeting of PEGylated medicines by blocking engager binding to PEG on the nanomedicines[25,26]. We measured anti-PEG antibody concentrations in healthy human plasma samples using an anti-PEG enzyme-linked immunosorbent assay (ELISA) developed in our lab (Supplementary Fig. 7a). The pre-existing anti-PEG IgG concentrations ranged from 0.3 to 237.5 µg ml$^{-1}$ with a mean concentration of $5.75 \pm 16.0$ µg ml$^{-1}$ in 386 anti-PEG positive samples[27]. We selected a human serum sample containing a relatively high concentration of anti-PEG IgG (51.4 µg ml$^{-1}$). However, PEG engager[EGFR] plus Doxisome in 20% human serum that was positive for anti-PEG IgG antibodies or in control human serum displayed similar EC$_{50}$ values against MDA-MB-468 cells (Supplementary Fig. 7b). These results suggest that pre-existing anti-PEG antibodies in patients do not effectively compete with PEG engager[EGFR], presumably due to the high anti-PEG affinity of the PEG engager and abundant PEG chains on Doxisome.

**Pharmacokinetics and tumour targeting of the PEG engager.** Pre-targeting of PEG engagers to tumours may allow for sub-sequent accumulation and endocytosis of PEGylated nanocarriers in cancer cells. The *in vivo* pharmacokinetics of PEG engagers was examined to determine a reasonable time point for administration of PEGylated nanocarriers after administration of PEG engager. The half-lives of the PEG engagers were $\sim 2.1$ h (PEG engager[EGFR]) and 2.2 h (PEG engager[CD19]) (Fig. 6a) after intravenous administration of 150 µg of PEG engager. Almost 90% of the PEG engagers were cleared from the circulation by 5 h after injection (Fig. 6a).

To visualize whether pre-targeting can facilitate the uptake and retention of PEGylated compounds in tumours, mice bearing established high EGFR expression levels (MDA-MB-468 and A431) or low EGFR expression levels (HepG2) tumours were intravenously injected with 6 mg kg$^{-1}$ PEG engager and then subsequently intravenously injected with 4armPEG$_{10k}$-NIR-797 probe 5 h later. *In vivo* imaging system (IVIS) optical imaging of these mice at 24, 48 and 72 h after probe injection showed that the fluorescence signal in PEG engager[EGFR] targeted tumours was significantly enhanced as compared with the PEG engager[CD19] control group (Fig. 6b and Supplementary Fig. 8). The fluorescent intensity in PEG engager[EGFR] targeted tumours at 24, 48 and 72 h was 2.7-fold, 2.1-fold and 2.8-fold greater than in the control PEG

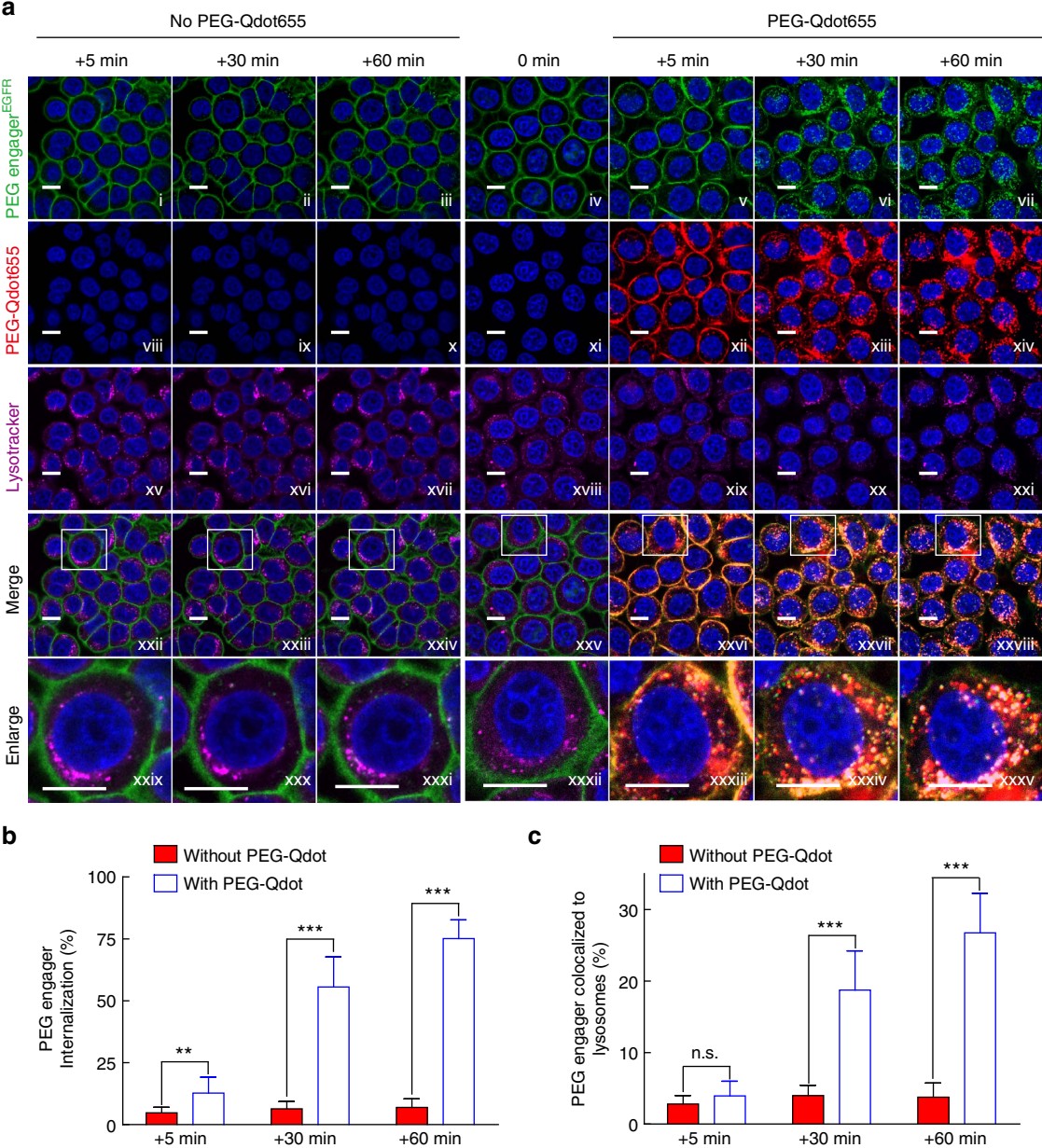

**Figure 4 | PEG engager$^{EGFR}$ conditional internalization. (a)** Fluorescent-labelled PEG engager$^{EGFR}$ on the cell membrane of MDA-MB-468 cells after 1 h at 37 °C (green, iv) was real-time imaged 5, 30 and 60 min after incubating without (i–iii) or with (v–vii) PEG-Qdot655. Hoechst 33342 (blue), PEG-Qdot655 (red, xiii–xiv) and LysoTracker Red DND-99 (purple pseudo colour, xv–xxi) are also shown for nucleus, Qdots and lysosome staining, respectively. Merged (xxii–xxviii) and enlarged (xxix–xxxv) images are also shown. Scale bar, 10 μm. **(b)** Percentage of the PEG engager$^{EGFR}$ that internalized into cells complexed with (white bars) or without (red bars) PEG-Qdot655 at different times was quantified from confocal images of individual cells ($n = 15$). **(c)** Percentage of PEG engager$^{EGFR}$ that co-localized with LysoTracker Red DND-99 (lysosomes) at different times was quantified from confocal images of individual cells ($n = 15$). Representative confocal images from two independent experiments are shown. Data are shown as mean ± s.d. Significant differences in percentage of PEG engager$^{EGFR}$ internalization or lysosome co-localization with and without addition of PEG-Qdot655 are indicated: **$P \leq 0.001$, ***$P \leq 0.0001$ (two-way analysis of variance). NS, not significant.

engager$^{CD19}$-treated tumours, respectively (Fig. 6c). Neither PEG engager$^{EGFR}$ nor PEG engager$^{CD19}$ enhanced the fluorescence signal in HepG2 (low EGFR expression levels) tumour-bearing mice (Supplementary Fig. 9).

**Antitumour activity of pre-targeted PEG engager.** To investigate whether PEG engager$^{EGFR}$ can inhibit EGFR signalling, EGFR-positive A431 cells were stimulated with or without epidermal growth factor and then co-incubated with PEG engagers

or control antibodies. Both Erbitux (monoclonal anti-EGFR IgG) and PEG engager$^{EGFR}$ inhibited the phosphorylation of EGFR and Erk as compared with negative control Herceptin (anti-HER2 IgG) and PEG engager$^{CD19}$ (Supplementary Fig. 10). Mice bearing human MDA-MB-468 or MDA-MB-231 TNBC xenografts were intravenously injected with phosphate-buffered saline (PBS), free doxorubicin (3 mg kg$^{-1}$) or PEG engager$^{CD19}$ or PEG engager$^{EGFR}$ (6 or 18 mg kg$^{-1}$) 5 h before Doxisome treatment (1 or 3 mg kg$^{-1}$) on days 1, 8, 15 and 22. Although engager$^{EGFR}$ inhibited EGFR signalling *in vitro*, mice treated only with PEG

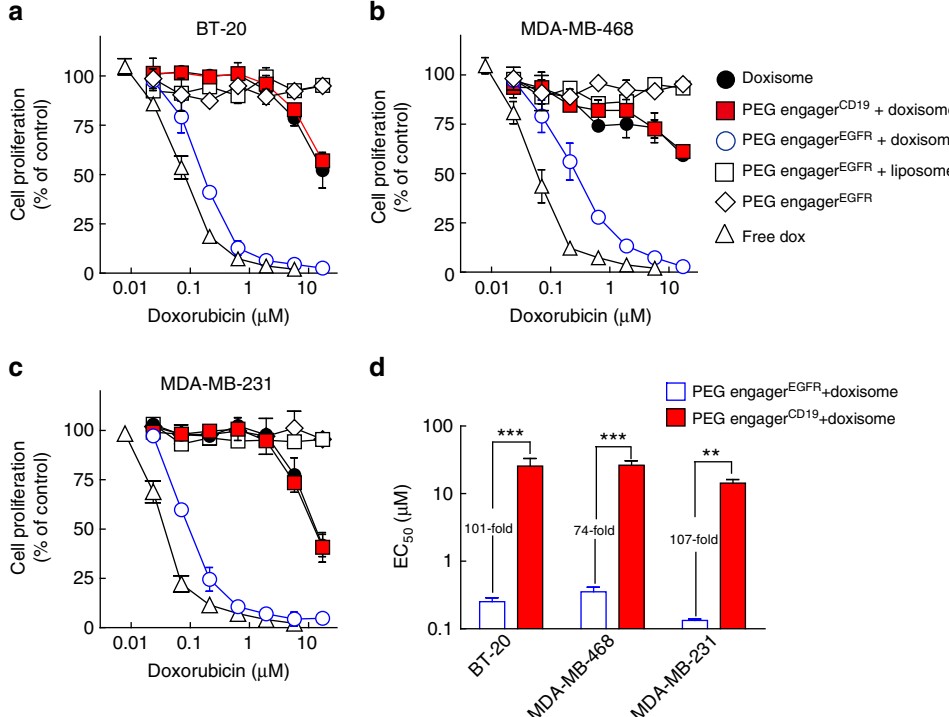

**Figure 5 | PEG engager$^{EGFR}$ enhances nanomedicine anti-proliferative activity.** BT-20 (**a**), MDA-MB-468 (**b**) and MDA-MB-231 cells (**c**) were incubated with PEG engager$^{EGFR}$ for 30 min followed by addition of serial dilutions of Doxisome (white circles) or empty liposomes (white squares) in triplicate for 4 h. The cells were also incubated with culture medium or PEG engager$^{CD19}$ for 30 min followed by addition of serial dilutions of Doxisome (black circles and red squares, respectively). Serial dilutions of free doxorubicin (white triangles) or PEG engager$^{EGFR}$ alone (white diamonds) were also added to cells for 4 h. The incorporation of $^{3}$H-thymidine into cellular DNA was measured 72 h later. The data are representative of three independent experiments. (**d**) The half maximal effective concentration (EC$_{50}$) values of BT-20, MDA-MB-468 and MDA-MB-231 cells treated with PEG engager$^{CD19}$ plus Doxisome or PEG engager$^{EGFR}$ plus Doxisome were analysed ($n = 3$). Data are shown as mean ± s.d. Significant differences in mean EC$_{50}$ values are indicated: **$P \leq 0.001$, ***$P \leq 0.0001$ (two-way analysis of variance).

engager$^{EGFR}$ displayed similar tumour growth as mice treated with PBS (Fig. 7a,c). Free doxorubicin suppressed tumour growth as compared to treatment of mice with PBS (Fig. 7a, $P < 0.05$). PEG engager$^{CD19}$ combined with 1 mg kg$^{-1}$ Doxisome or 1 mg kg$^{-1}$ Doxisome alone displayed similar and better suppression of tumour growth as compared to treatment of mice with free doxorubicin or PBS vehicle (Fig. 7a, $P < 0.0001$ and Fig. 7c, $P < 0.001$). PEG engager$^{EGFR}$ plus Doxisome significantly suppressed TNBC tumour growth as compared to mice treated with Doxisome alone (Fig. 7a, $P < 0.005$ and Fig. 7c, $P = 0.0012$). The maximum-tolerated dose of doxorubicin in severe combined immunodeficiency (SCID) mice is around 2.5–3 mg kg$^{-1}$ due to defective DNA repair in these mice[28]. Increasing the dose of Doxisome to 3 mg kg$^{-1}$ did not provide better therapeutic activity because the mice experienced significant body weight loss and early deaths (Fig. 7b). Our results demonstrate that pre-targeting PEG engager$^{EGFR}$ to EGFR overexpressing TNBC tumours can markedly enhance the therapeutic efficacy of PEGylated liposomal doxorubicin (Doxisome) with limited side effects as shown by body weight loss analysis (Fig. 7b).

We further investigated whether pre-docking of PEGylated nanoparticles with PEG engagers could enhance their therapeutic efficacy. We used a molar ratio of PEG engager and PEG-lipid on Doxisome of 1:55 (Supplementary Fig. 11a). Mice were administrated with a mixture of PEG engager and Doxisome (pre-incubated at 4 °C for 1 h) and blood samples were then periodically collected from the tail vein of the mice. The half-lives of the PEG engagers as determined by quantitative ELISA were ~3.5 h (PEG engager$^{EGFR}$) and 3.8 h (PEG engager$^{CD19}$) (Supplementary Fig. 11b) after intravenous administration of

PEG engager-docked Doxisome (containing 30 μg PEG engager). To examine the therapeutic activity of PEG engager-docked Doxisome, non-obese diabetic-severe combined immunodeficiency (NOD SCID) mice bearing human MDA-MB-468 TNBC xenografts were intravenously injected with PBS, 3 mg kg$^{-1}$ free doxorubicin, 6 mg kg$^{-1}$ PEG engager$^{EGFR}$ alone, PEG engager$^{CD19}$-decorated Doxisome or PEG engager$^{EGFR}$-decorated Doxisome (1 mg kg$^{-1}$ of doxorubicin) on days 1, 8, 15 and 22. Doxorubicin slightly inhibited tumour growth while significantly better antitumour activity was observed in mice treated with Doxisome (1 mg kg$^{-1}$) or PEG engager$^{CD19}$-decorated Doxisome as compared to mice treated with PBS vehicle (Supplementary Fig. 11c, $P < 0.005$). A higher dose (3 mg kg$^{-1}$) of Doxisome was toxic to the mice (Fig. 7b and Supplementary Fig. 11d). PEG engager$^{EGFR}$-decorated Doxisome significantly suppressed TNBC tumour growth as compared to mice treated with Doxisome alone (Supplementary Fig. 11c, $P < 0.05$). Thus, pre-docking PEG engager$^{EGFR}$ on Doxisome markedly enhanced therapeutic efficacy with minimal side effects as measured by body weight loss (Supplementary Fig. 11d).

**Off-target effects of PEG engager-mediated therapy.** EGFR is expressed at low levels on normal cells of epithelial, mesenchymal and neuronal origin. We hypothesized that the density of EGFRs on cells might be an important factor for conditional internalization of PEGylated nanocarriers by PEG engager$^{EGFR}$ (Fig. 8a). Indeed, a linear correlation was observed between the logarithm of EGFR expression levels on cancer cell lines and the logarithm of the anti-proliferative activity (EC$_{50}$ values) of PEG

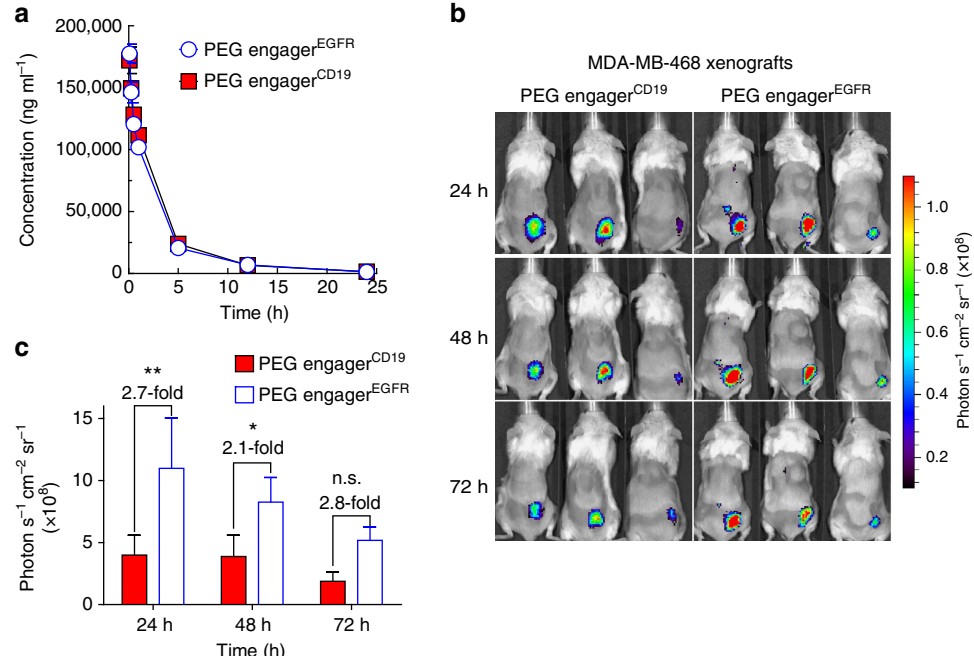

**Figure 6 | Pharmacokinetics and imaging of PEG engagers in mice. (a)** NSG mice were intravenously injected with 6 mg kg$^{-1}$ PEG engager$^{EGFR}$ (white circles) or PEG engager$^{CD19}$ (red squares). Mean plasma concentrations of the PEG engagers were measured by sandwich ELISA ($n=3$ mice). **(b)** Five hours before intravenous administration of 4armPEG$_{10k}$-NIR-797 probe (5 mg kg$^{-1}$), NSG mice bearing subcutaneous MDA-MB-468 tumours were intravenously injected with 6 mg kg$^{-1}$ PEG engager$^{EGFR}$ or PEG engager$^{CD19}$ and the whole-body imaging were sequentially imaged at 24, 48 and 72 h with an IVIS Spectrum imaging system. **(c)** The uptake of PEG-NIR797 in MDA-MB-468 tumours was determined by measuring fluorescence intensities ($n=3$). Data are shown as mean ± s.d. Significant differences in mean fluorescent intensity between PEG engager$^{EGFR}$ and PEG engager$^{CD19}$ groups are indicated: $*P \leq 0.01$, $**P \leq 0.001$ (two-way analysis of variance). NS, not significant.

engager$^{EGFR}$ plus Doxisome treatment (Fig. 8b; $R^2 = 0.702$). It has been reported that EGFR-targeted therapies can cause hepatotoxicity[29]. EGFR expression in normal human hepatocytes (mean fluorescence intensity = 38), however, is relatively low (Fig. 8c), which suggests reduced off-target toxicity by PEG engager$^{EGFR}$ therapy.

## Discussion

We report bispecific PEG engagers that simultaneously bind to PEG on nanomedicines and EGFR on cancer cells for selective delivery of PEGylated stealth nanocarriers to EGFR$^+$ TNBC cells. PEG engager$^{EGFR}$ bound to EGFRs on TNBC cells but remained in a dormant state on the plasma membrane until contact with PEGylated nanocarriers triggered rapid endocytosis of both the engager and PEGylated nanocarrier. PEG engager$^{EGFR}$ markedly increased the anticancer activity of PEG-liposomal doxorubicin (Doxisome) against EGFR$^+$ TNBC *in vitro* and *in vivo*. The PEG engager tolerated the presence of pre-existing anti-PEG antibodies and effectiveness correlated to the EGFR expression levels on cells. This simple and flexible strategy appears promising for enhanced delivery of PEGylated medicines for improved therapy of TNBC patients.

TNBC is highly aggressive and metastatic with high rates of recurrence[2]. About 50% of TNBC tumours overexpress EGFR, which is correlated with poor prognosis in TNBC patients[3]. Overexpressed EGFR can induce cell proliferation, promote angiogenesis, increase cell migration and enhance chemoresistance[30]. Therefore, EGFR has been considered as a therapeutic target for the treatment of TNBC. However, several receptor tyrosine kinase inhibitors (RTKIs) including gefitinib and erlotinib did not benefit TNBC patients[5,6]. Compared to wild-type EGFR, EGFRs with activating mutations in non-small

cell lung cancers display higher binding to RTKIs, leading to 100-fold greater sensitivity to RTKIs[31]. Thus, the poor efficacy of RTKIs in TNBC may be due to the low occurrence of activating EGFR mutations in TNBC (3–11%)[32,33] as compared to non-small cell lung cancers (10–35%)[34,35]. In contrast to RTKIs that target activated intracellular EGFR kinase domains, antibodies that recognize the extracellular domain of the EGFR can target both wild-type and mutated EGFRs (Fig. 5 and Supplementary Fig. 4). Therapeutic monoclonal antibodies that recognize extracellular domains of EGFR, such as cetuximab, have been applied for TNBC therapy. However, the response rate of cetuximab alone or combined with carboplatin for TNBC treatment was not promising[36]. These EGFR-targeted antibodies and RTKIs focus on the blockage of EGFR signalling pathways, which involves complicated signalling networks. For instance, temsirolimus can inhibit the EGFR-PI3K-AKT-mTOR pathway resulting in cancer cell death. However, MDA-MB-231 TNBC cells, which possess normal phosphatase and tensin homologue function (a tumour suppressor gene that inactivates AKT) are resistant to temsirolimus, whereas MDA-MB-468 TNBC cells (with loss of phosphatase and tensin homologue function) were 8,000-fold more sensitive to this drug[37]. Alternatively, delivery of anticancer drugs to TNBC cells by targeting EGFRs may bypass the drawbacks of EGFR signalling inhibition. Thus, PEG engager$^{EGFR}$ may be useful for selective delivery of PEGylated nanomedicines to EGFR overexpressing TNBC.

Active targeting of nanomedicines to tumours by functionalizing nanocarriers with targeting ligands is attractive to improve tumour targeting and intracellular delivery of nanocargos[13,14]. However, the complexity involved in production of targeted nanomedicines is time-consuming, technically challenging and expensive. The technical challenges include difficulty in controlling ligand conjugation stoichiometry, aggregation of

ligands on nanocarrier surfaces, intercarrier variations in ligand density, loss of ligand bioactivity, deattachment of targeting ligands from nanocarriers *in vivo* and poor scalability or reproducibility of the conjugation process[17]. The resulting batch-to-batch variations can hinder clinical translation and commercialization of targeted nanocarriers. To help overcome these problems, Brinkmann and colleagues developed bispecific digoxigenin-binding antibodies for delivery of digoxigen-modified nanocarriers to disease sites[38,39]. Likewise, we previously described bispecific PEG-binding antibodies for tumour-specific delivery of PEGylated compounds[40,41]. However, direct attachment of targeting ligands to nanocarriers can reduce the stealth feature of the nanocarriers, resulting in accelerated clearance and reduced uptake into tumours[10]. In addition, attachment of targeting ligands can increase nanocarrier size, further impeding tumour uptake[18].

Pre-targeting strategies may help translate targeted nanomedicines to the clinic by decreasing nanocarrier complexity,

maintaining the stealth properties of nanocarriers and facilitating personalized combinations of targeting ligands and nanomedicines. For example, monovalent ligands can be targeted to high density receptors on cancer cells for subsequent internalization of nanocarriers modified with biorthogonal click-chemistry agents[42,43]. However, both the targeting ligand and the nanocarrier require chemical modification and the kinetics of the biorthogonal click-chemistry reactions may limit nanocarrier capture *in vivo*[44]. By contrast, PEG engagers can be used with any nanomedicine that is coated with PEG, which is commonly used to decrease nanocarrier uptake in macrophages and increase nanocarrier circulation time. Thus, even 'off-the-shelf' nanomedicines, such as Doxil, can be targeted to EGFR$^+$ TNBC tumours by this approach. Antibody–antigen interactions also rapidly occur *in vivo*, promoting effective capture and internalization of stealth nanocarriers.

Conditional internalization of nanocarriers into target cells is desirable for successful pre-targeted delivery[43,45]. Receptor dimerization stimulates EGFR endocytosis for turnover and downregulation of EGFRs[46,47]. Anti-EGFR monoclonal antibodies, such as cetuximab and necitumumab, recognize the domain III ligand-binding region of EGFR and rapidly trigger crosslinking of EGFRs due to their bivalent format, thereby stimulating internalization and degradation of the receptor[48]. Since bivalent antibodies are typically required to crosslink receptors and trigger endocytosis, we designed a monovalent PEG engager$^{EGFR}$ to accumulate at EGFRs without promoting receptor internalization. PEG engagers were retained on target cancer cells until contact with PEGylated nanocarriers, which crosslinked EGFRs and induced rapid endocytosis of the nanomedicines.

Doxil, PEGylated liposomal doxorubicin, is approved by FDA for treatment of ovarian and breast cancer patients[9,49]. Although the standard dose of Doxil recommended by the FDA (50 mg m$^{-2}$) significantly benefits cancer patients, a high incidence of hand-foot syndrome is also observed[9,50]. Doxil treatment is often delayed for patients with hand-foot syndrome, limiting its effectiveness. PEG engager$^{EGFR}$ enhanced the antitumour activity of low-dose Doxisome (1 mg kg$^{-1}$ = 3 mg m$^{-2}$), to TNBC tumours[51]. This significant tumour suppression may be due to the specific targeting and rapid endocytosis of Doxisome via pre-targeted PEG engagers. Therefore, the PEG engager-mediated liposomal doxorubicin therapy may allow effective therapy at lower doses, thus reducing side effects such as hand-foot syndrome. We also demonstrated

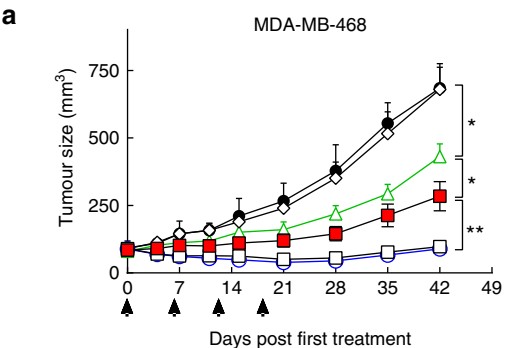

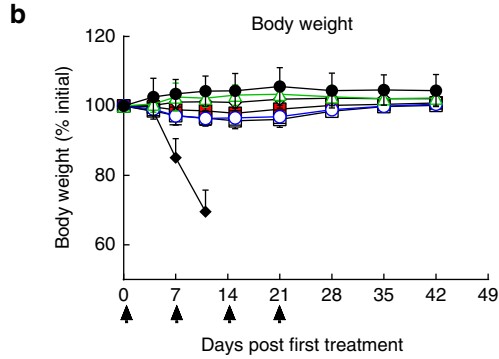

- ●-PBS
- ◆ LD 3 mg kg$^{-1}$
- △ 6 mg kg$^{-1}$ free doxorubicin
- ◇ 18 mg kg$^{-1}$ PEG engager$^{EGFR}$
- ■ 6 mg kg$^{-1}$ PEG engager$^{CD19}$ +LD 1 mg kg$^{-1}$
- □ 18 mg kg$^{-1}$ PEG engage$^{EGFR}$ +LD 1 mg kg$^{-1}$
- ○ 6 mg kg$^{-1}$ PEG engager$^{EGFR}$ +LD 1 mg kg$^{-1}$

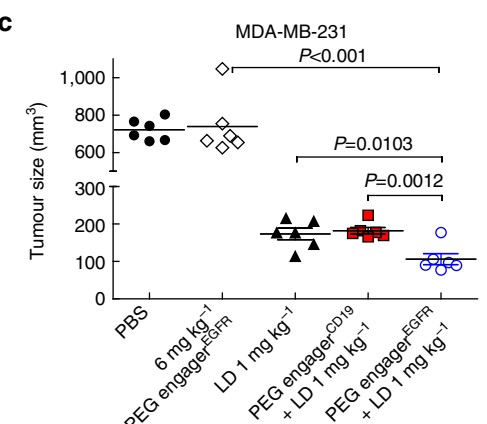

**Figure 7 | Therapeutic efficiency of PEG engager-directed Doxisome.**
(**a**) Groups of eight SCID mice bearing MDA-MB-468 tumours were intravenously injected with 6 mg kg$^{-1}$ PEG engager$^{CD19}$ (red squares), 6 mg kg$^{-1}$ PEG engager$^{EGFR}$ (white circles) or 18 mg kg$^{-1}$ PEG engager$^{EGFR}$ (white squares) 5 h before intravenous injection of 1 mg kg$^{-1}$ Doxisome. Groups of eight mice were also intravenously injected with 6 mg kg$^{-1}$ free doxorubicin (white triangles), 3 mg kg$^{-1}$ Doxisome (black diamonds), PBS (black circles) or 18 mg kg$^{-1}$ PEG engager$^{EGFR}$ (white diamonds). Treatment was performed once a week for 4 weeks (arrows). Results show mean tumour sizes ± s.d. (**b**) Mean body weights of treated MDA-MB-468 mice ($n = 8$). (**c**) Groups of six SCID mice bearing MDA-MB-231 tumours were intravenously injected with 6 mg kg$^{-1}$ PEG engager$^{EGFR}$ (white circles) or 6 mg kg$^{-1}$ PEG engager$^{CD19}$ (red squares) 5 h before intravenous injection of 1 mg kg$^{-1}$ Doxisome. Groups of six mice were also intravenously injected with PBS (black circles), PEG engager$^{EGFR}$ alone (white diamonds) or 1 mg kg$^{-1}$ Doxisome (black triangles). Treatment was performed once a week for 4 weeks. Results show mean tumour sizes at 43 days post first treatment). Statistical analysis of the differences in tumour volumes between treatment and control groups was performed by one-way analysis of variance (ANOVA) followed by Dunnett's multiple comparisons. *$P \leq 0.05$, **$P \leq 0.005$.

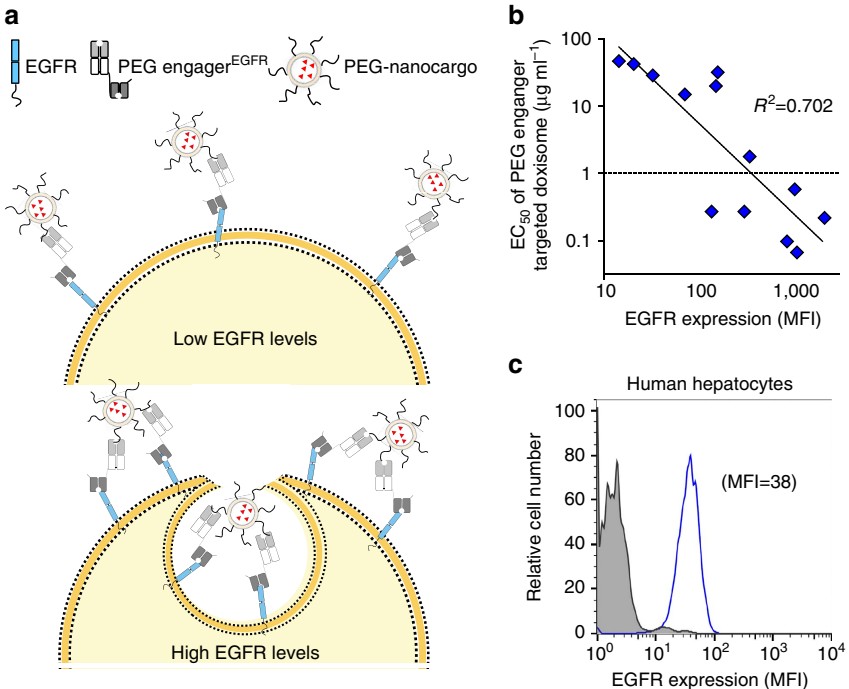

**Figure 8 | EGFR levels correlate with PEG engager-directed Doxisome therapy.** (**a**) The proposed mechanism of PEG engager-mediated internalization of PEGylated nanoparticles. (**b**) Drug sensitivity to PEG engager-targeted Doxisome ($EC_{50}$ values) is plotted against EGFR expression levels on cancer cell lines. (**c**) Live human hepatocytes were immunofluorescence stained with control antibodies (grey area) or anti-human EGFR antibodies (blue solid line) and then analysed on a flow cytometer. Mean fluorescence intensity (MFI). Half maximal effective concentration ($EC_{50}$).

that pre-decoration of Doxisome with PEG engager[EGFR] could enhance the half-life of the engagers and improve the therapeutic index of Doxisome treatment against EGFR$^+$ human tumour xenografts, indicating that both pre-mixing and pre-targeting of PEG engagers are promising therapeutic approaches.

Stepwise accumulation of mutations can lead to distinct cancer cell subpopulations within the same tumour[52,53]. For example, discordant HER2 expression has been observed in the same biopsy from cancer patients[54]. This could be a major obstacle for conventional single-ligand targeted cancer therapy since subpopulations lacking expression of target receptors may repopulate the tumour after treatment. To overcome this problem, multi-ligand nanocarriers can be employed to improve drug delivery to heterogeneous tumours[55,56]. A cocktail of PEG engagers with specificity to different tumour-associated antigens (for example, insulin-like growth factor 1 receptor, HER2, HER3, HER4 and c-Met) might offer a flexible yet effective treatment option for heterogeneous tumours.

Antibody-drug conjugates (ADC) are one of the most promising targeted drug delivery systems. ADCs are composed of monoclonal antibodies conjugated with cytotoxic drug molecules via a biodegradable linker to allow drug release after ADC internalization into cancer cells[57]. Low drug-to-antibody ratios of two–four are required to prevent alteration of antibody properties and maintain efficient cancer therapy, thereby requiring attachment of highly cytotoxic drugs such as monomethyl auristatin E and maytansine[58–60]. EGFR is also a potential target for developing ADCs as two EGFR-targeting ADCs are in clinical trials (ABT-414/Abbvie[61] and IMGN289/ ImmunoGen). However, off-target uptake of ADCs in normal cells which express low levels of the target receptors can cause severe side effects[62]. We determined that EGFR expression density is critical for efficient PEG engager[EGFR]-mediated endocytosis of PEGylated liposomal doxorubicin, suggesting that PEG engagers may possess lower off-target toxicity.

This may expand the range of targetable tumour antigens as well as allow delivery of a broader range of therapeutic cancer drugs since each nanocarrier can encapsulate up to 10,000 drug molecules[63]. On the other hand, PEG engagers might be less effective for treatment of cancers that express low levels of target antigens.

Covalent attachment of PEG to peptides, proteins, nucleic acids and nanoparticles can improve their pharmacokinetic properties and stability[64]. Although PEG is thought to be non-immunogenic, several studies have detected pre-existing anti-PEG antibodies in normal donors[26,65,66]. Pre-existing anti-PEG antibodies could theoretically compromise the effectiveness of PEG engagers by blocking engager binding to PEGylated medicines. However, we found that human serum that contained high levels of anti-PEG antibodies did not interfere with PEG engager-mediated delivery of Doxisome into TNBC cells. This may be because PEG engagers display high affinity to PEG ($K_D = 7.55 \, nM$) and are not easily competed by naturally occurring anti-PEG antibodies.

In conclusion, the humanized PEG engager generated in this study is anticipated to help overcome major bottlenecks in targeted nanomedicines by reducing manufacturing complexity and providing flexibility in choosing the appropriate disease targets by simply changing the targeting portion of these molecules. Flexible targeting coupled with the capability to directly deliver nanocarriers into target cells may expand the range of therapeutic agents available for therapy.

## Methods

**Cell lines and animals.** Human 293FT cells (Thermo Fisher Scientific, San Jose, CA) were cultured in Dulbecco's modified Eagle's medium (Sigma-Aldrich, St Louis, MO) supplemented with 10% heat-inactivated foetal calf serum (HyClone, Logan, Utah), 100 U ml$^{-1}$ penicillin and 100 µg ml$^{-1}$ streptomycin at 37 °C in an atmosphere of 5% $CO_2$ in air. Human hepatocytes were purchased from BD Biosciences (San Jose, CA) EGFR-mutated PC9 (EGFR exon19del E746-A750) cell line was kindly provided by Dr Pan-Chyr Yang (President of the National Taiwan

University, Taiwan). PC3, SKBR3, SK-MES-1, Hut125, Caski, HT29, H2170, LS174T, HepG2, SW480, MCF7 and TNBC cell lines (MDA-MB-231, MDA-MB-468 and BT-20 cells) were obtained from American Type Culture Collection (Manassas, VA) and were maintained in RPMI-1640 containing the same supplements. Knockdown of EGFR in BT20 cells is described in Supplementary Methods. Except for BT-20 cells, the cell lines used in this study are not listed in the database of commonly misidentified cell lines maintained by the International Cell Line Authentication Committee. The cell lines were not authenticated by our laboratory. All cell lines were tested for mycoplasma and propagated for less than 6 months after resuscitation. Female NSG mice (non-obese diabetic. Cg-Prkdc$^{scid}$ Il2rg$^{tm1Wjl}$/SzJ, 6–8 week old), NOD SCID mice (NOD.CB17-Prkdc$^{scid}$/NcrCrl, 6–8 week old) and BALB/c nude mice (BALB/cAnN.Cg-Foxn1$^{nu}$/CrlNarl) were obtained from the National Laboratory Animal Center, Taipei, Taiwan and were maintained under specific pathogen-free conditions. All animal experiments were performed in accordance with institutional guidelines and ethically approved by the Laboratory Animal Facility and Pathology Core Committee of IBMS, Academia Sinica. Mice were randomly assigned to treatment groups but the investigators were not blinded to the treatments.

**DNA plasmid construction.** pAS3w.Ppuro, pMD.G (VSV-G envelope plasmid) and pCMVΔR8.91 (packaging plasmid) vectors were obtained from the National RNAi Core Facility (Institute of Molecular Biology/Genomic Research Center, Academia Sinica, Taiwan)[67]. To generate the anti-PEG Fab-based bispecific PEG engager antibodies, the mouse $V_L$ and $V_H$ domains of the 6.3 antibody were cloned from cDNA prepared from the 6.3 hybridoma[40]. The anti-PEG antibody was humanized by first aligning the $V_H$ and $V_L$ sequences of the mouse 6.3 antibody to human immunoglobulin germline sequences using the IgBLAST program (http://www.ncbi.nlm.nih.gov/igblast/). The human germline $V_H$ IGHV7-4-1*02 and $V_L$ IGKV4-1*01 exons were selected based on the framework homology. The complementarity-determining regions of mouse 6.3 $V_H$ and $V_L$ domains were then grafted onto human $V_H$ IGHV7-4-1*02 and $V_L$ IGKV4-1*01 genes using assembly PCR. Human immunoglobulin $G_1$ (IgG$_1$) $C_κ$ and CH$_1$ constant domains were cloned from extracted human peripheral blood mononuclear cells cDNA. Humanized 6.3 $V_L$–$C_κ$ and 6.3 $V_H$–CH$_1$ domains were assembled by overlap polymerase chain reaction from humanized 6.3 $V_L$ and 6.3 $V_H$ and human $C_κ$ and CH$_1$ fragments[41]. The humanized 6.3 $V_L$–$C_κ$ and 6.3 $V_H$–CH$_1$ were joined by a composite internal ribosome entry site bicistronic expression peptide linker and inserted into the pAS3w.Ppuro plasmid. The hBU12 (anti-human CD19) and Necitumumab (IMC-11F8, anti-human EGFR) single chain dsFv were synthesized by assembly PCR based on the $V_H$ and $V_L$ sequences of hBU12 and Necitumumab from US patents US7968687B2 and US7598350B2, respectively. The dsFv DNA fragments were digested with MfeI I and Mlu I and then subcloned into the pAS3w.Ppuro-PEG engager plasmid to generate pAS3w.Ppuro-PEG engager$^{CD19}$ and pAS3w.Ppuro-PEG engager$^{EGFR}$.

**Production of recombinant bispecific PEG engager antibodies.** 293FT/PEG engager$^{CD19}$ and 293FT/PEG engager$^{EGFR}$ cells that stably secreted PEG engager$^{CD19}$ and PEG engager$^{EGFR}$ were generated by lentiviral transduction[67]. Recombinant lentiviral particles were packaged by co-transfection of pAS3w.Ppuro-pAS3w.Ppuro-PEG engager$^{CD19}$ and pAS3w.Ppuro-PEG engager$^{EGFR}$ (7.5 μg) with pCMVΔR8.91 (6.75 μg) and pMD.G (0.75 μg) using TransIT-LT1 transfection reagent (Mirus Bio) (45 μl) in 293FT cells grown in a 10 cm culture dish (90% confluency). After 48 h, lentiviral particles were collected and concentrated by ultracentrifugation (Beckman SW 41 Ti Ultracentrifuge Swing Bucket Rotor, 50,000g, 1.5 h, 4 °C). Lentiviral particles were suspended in culture medium containing 5 μg ml$^{-1}$ polybrene and filtered through a 0.45 μm filter. 293FT cells were seeded in six-well plates ($1 \times 10^5$ cells per well) 1 day before viral infection. Lentivirus containing medium was added to the cells and then centrifuged for 1.5 h (500g, 32 °C). The cells were selected in puromycin (5 μg ml$^{-1}$) to generate stable cell lines. $5 \times 10^7$ 293FT/PEG engager$^{CD19}$ or 293FT/PEG engager$^{EGFR}$ cells in 15 ml DMEM culture medium were cultured in CELLine adhere 1000 bioreactors (INTEGRA Biosciences AG) and the medium was collected from every 7–10 days. Polyhistidine-tagged bispecific antibodies were purified on a Co$^{2+}$-TALON column (GE Healthcare Life Sciences). Protein concentrations were determined by the bicinchoninic acid protein assay (Thermo Fisher Scientific).

**Characterization of PEG engagers.** Three microgram of Erbitux, purified PEG engager$^{CD19}$ or PEG engager$^{EGFR}$ were electrophoresed in a 10% SDS–PAGE gel under reducing or non-reducing conditions and then stained by Coomassie blue. For further analytical characterization, the antibodies were dialysed into water and analysed by matrix-assisted laser desorption/ionization time-of-flight mass spectrometry (New ultrafleXtreme, Bruker). Additional characterization of PEG engagers is descibed in Supplementary Methods.

**Fluorescent labelling of PEG engager antibodies.** An amount of 5 mg of purified PEG engager$^{CD19}$ or PEG engager$^{EGFR}$ antibodies in coupling buffer (0.1 M sodium bicarbonate, pH = 8.0) was mixed with a 10-fold molar excess of Alexa Fluor 647

succinimidyl esters (Thermo Fisher Scientific) (in dimethyl sulfoxide) for 2 h at room temperature to produce Alexa Fluor 647-conjugated PEG engager$^{CD19}$ or PEG engager$^{EGFR}$, respectively. One-tenth volume of 1 M glycine solution (pH = 8.0) was added to stop the reaction. The PEG engagers were dialysed (molecular weight cutoff ∼ 12,000–14,000 daltons) against PBS to remove free Alexa Fluor 647, sterile filtered and stored at − 80 °C.

**Microscale thermophoresis-binding analysis.** HEPES buffered saline/CHAPS buffer (10 mM HEPES, 150 mM NaCl, 3 mM EDTA, 0.05% CHAPS, pH = 7.4) was used for sample preparation. To determine the PEG-binding affinity of PEG engagers, 5 nM of Cy5-conjugated methoxy PEG$_{5k}$ (Nanocs) was mixed at a 1:1 volume ratio with graded concentrations (0.24–500 nM) of PEG engager$^{CD19}$ or PEG engager$^{EGFR}$ antibodies. To analyse the tumour antigen-binding affinity of PEG engagers, 2 nM of Alexa Fluor 647-conjugated PEG engager$^{CD19}$ or PEG engager$^{EGFR}$ were mixed at a 1:2 volume ratio with graded concentrations (0.027–180 nM) of recombinant CD19 or EGFR proteins (Sino Biological Inc.). The samples were incubated for 5 min at room temperature and loaded into standard capillaries and heated at 5% LED and 40% laser power for 30 s, cooled for 10 s and measured on a NanoTemper Monolith NT.115 instrument (NanoTemper Technologies GmbH). All experiments were performed with three replicates.

**Flow cytometer analysis.** Surface expression of EGFR in human hepatocytes, PC3, SKBR3, SK-MES-1, Hut125, Caski, HT29, H2170, LS174T, HepG2, SW480, MDA-MB-231, MDA-MB-468 and BT-20 cells was determined by staining the cells with monoclonal mouse IgG anti-human EGFR (Santa Cruz Biotechnology, 5 μg ml$^{-1}$ in staining buffer (PBS containing 0.1% bovine serum albumin) for 30 min at 4 °C followed by Alexa Fluor 647-conjugated goat Ig anti-mouse IgG antibody (Thermo Fisher Scientific, 5 μg ml$^{-1}$). Unbound antibodies were removed by washing with cold PBS twice and the surface fluorescence of $10^4$ viable cells was measured by FACScaliber flow cytometer (Becton Dickinson) and analysed with Flowjo (Tree Star Inc.).

**Confocal microscopy.** Coverslips (30 mm) in cell cultivation POCmini (perfusion, open and closed) chambers (PeCon GmbH) were coated with 10 μg ml$^{-1}$ poly-L-lysine (Sigma-Aldrich) in PBS for 30 min at room temperature. The coverslips were washed twice with PBS and then $5 \times 10^4$ MDA-MB-468 (EGFR+), A431 (EGFR+), BT-20 (EGFR+) or MCF7 (EGFR−) cancer cells were seeded on the coverslips. Cancer cell-specific uptake of PEGylated nanoparticles was examined by staining the cells with 10 μg ml$^{-1}$ of PEG engager$^{CD19}$ or PEG engager$^{EGFR}$ antibodies at 37 °C for 30 min in medium containing 1 μg ml$^{-1}$ of Hoechst 33342 (Thermo Fisher Scientific). Unbound PEG engagers were removed by washing with PBS twice and then 8 nM of PEGylated Qtracker 655 non-targeted quantum dots (PEG-Qdot655, Thermo Fisher Scientific) in medium (RPMI-1640, 10% FBS) were added to the cells, which were visualized by real-time imaging on an Axiovert 200M Confocal Microscope (Carl Ziess Inc.) at excitation and emission wavelengths of 350 and 461 nm for Hoechst 33342 and 350 and 675 nm for PEG-Qdot655 at 37 °C, 5% CO$_2$. Similarly, conditional internalization of PEGylated nanoparticles was determined by incubating MDA-MB-468 or BT-20 cells with 10 μg ml$^{-1}$ of Alexa Fluor 647-conjugated PEG engager$^{EGFR}$ (excitation/emission, 650 nm/675 nm) at 37 °C for 30 min in medium containing 1 μg ml$^{-1}$ of Hoechst 33342 and 100 nM of Lyso-Tracker Red DND-99 (appears in purple pseudo colour). After washing, the cells were incubated at 37 °C for 1 or 9 h and imaged on the Axiovert 200M Confocal Microscope, followed by real-time cell imaging after adding 8 nM of PEG-Qdot655 solution. The percentages of internalized PEG engagers and PEG-Qdots were calculated by dividing the fluorescence of the intracellular regions by the whole-cell fluorescence based on the bright field cell images using ZEN 2011 software (blue edition) (Carl Zeiss, Jena, Germany).

**Ethical statement.** The studies were approved by the Institutional Review Boards and Ethics Committees of Academia Sinica in Taiwan. Written informed consent was obtained from the subjects in accordance with institutional requirements and Declaration of Helsinki principles.

**Plasma sample collection.** Plasma samples of healthy subjects were enrolled from a prior project that had been collected, centrifuged and stored at the National Center for Genome Medicine, Academia Sinica. All subjects of this study agreed to offer the remaining centrifugal plasma for other research by treatment de-link. Details of anti-PEG antibody detection and their effects on PEG engagers is described in Supplementary Methods.

**PEGylated liposome preparation.** Distearoyl phosphatidylcholine, 1,2-distearoyl-sn-glycero-3-phosphoethanolamine-N-[methoxy(polyethylene glycol)-2000 (DSPE-PEG$_{2000}$) and cholesterol (Avanti Polar Lipids, Inc.) were dissolved in chloroform at a 65:5:30 molar ratio, respectively. A dried lipid film was formed at 65 °C by rotary evaporation (Buchi, Rotavapor RII) and rehydrated in Tris-buffered saline (TBS, 50 mM Tris-HCl, 150 mM NaCl, pH 7.4) at 65 °C to a final lipid concentration of 20 mg ml$^{-1}$. The liposomal suspension was

submitted to 10 freeze/thaw cycles in liquid nitrogen and a heated water bath at 80 °C, followed by 21 extrusions at 75 °C through 400, 200 and 100 nm polycarbonate membranes each using a mini-extruder (Avanti Polar Lipids, Inc.). The final lipid concentration was measured by Bartlett's assay[68] and adjusted to 13.9 µmol ml$^{-1}$ with TBS before use.

**Cell proliferation assay.** PC3, SKBR3, SK-MES-1, Hut125, Caski, HT29, H2170, LS174T, HepG2, SW480 MDA-MB-231, MDA-MB-468, BT-20 and PC9 cells (10,000 cells per well) were seeded in 96-well plates overnight. Serial dilutions of free doxorubicin or vinorelbine were directly added to the cells as positive controls. Fifteen microgram per ml of PEG engager$^{CD19}$ or PEG engager$^{EGFR}$ antibodies were added to the cells for 30 min at 37 °C followed by addition of graded concentrations of PEGylated liposomal doxorubicin (Doxisome, 13.9 µmol ml$^{-1}$ lipid concentration, Taiwan Liposome Company Ltd., Taipei, Taiwan) or liposomal vinorelbine[69] (provided by Dr Han-Chung Wu, Research Fellow, Institute of Cellular and Organismic Biology, Academia Sinica, Taiwan) or empty liposomes to the cells in triplicate at 37 °C for 4 h. The cells were subsequently washed once and incubated for an additional 72 h in fresh culture medium and then pulsed for 18 h with $^3$H-thymidine (1 µCi per well). Results are expressed as per cent inhibition of $^3$H-thymidine incorporation into cellular DNA in comparison to untreated cells.

**In vivo pharmacokinetics.** NSG mice were intravenously injected with 150 µg PEG engager$^{CD19}$ or PEG engager$^{EGFR}$ and blood samples were periodically collected from the tail vein of the mice. Plasma was prepared by centrifugation (5 min, 12,000g). The PEG engager levels in plasma were determined by quantitative sandwich ELISA. Maxisorp 96-well microplates were coated with 50 µl per well of anti-6 × His tag antibody (GeneTex) (2 µg ml$^{-1}$) in bicarbonate buffer, pH 8.0 for 4 h at 37 °C and then at 4 °C overnight. The plates were blocked with 200 µl per well 5% skim milk in PBS for 2 h at room temperature and then washed with PBS three times. Graded concentrations of PEG engager$^{CD19}$, PEG engager$^{EGFR}$ or plasma samples in dilution buffer (2% skim milk in PBS) were added to the wells for 2 h at room temperature. After washing with PBS four times, the plates were stained with 50 µl per well horseradish peroxidase-conjugated anti-human IgG Fab antibody (Jackson ImmunoResearch Laboratories) (5 µg ml$^{-1}$). The plates were washed with PBS six times and 100 µl per well ABTS solution (0.4 mg ml$^{-1}$ 2,2′-azino-di(3-ethylbenzthiazoline-6-sulfonic acid), 0.003% $H_2O_2$, 100 mM phosphate citrate, pH 4.0) was added for 30 min at room temperature. The absorbance of the wells at 405 nm was measured on a microplate reader. The initial and terminal half-lives of the PEG engagers were estimated by fitting the data to a two-phase exponential decay model with Prism 5 software (Graphpad Software).

**Synthesis of PEGylated near-infrared probes.** 4arm-PEG$_{10K}$-NH$_2$ (Laysan Bio) dissolved in dimethyl sulfoxide at 2 mg ml$^{-1}$ was mixed with a sixfold molar excess of NIR-797 isothiocyanate (Santa Cruz Biotechnology) (in dimethyl sulfoxide) for 2 h at room temperature to produce 4arm-PEG$_{10K}$- NIR-797 probes. These compounds were diluted in a fivefold volume of ddH$_2$O and dialysed (molecular weight cutoff ∼ 12,000–14,000 daltons) against ddH$_2$O to remove free NIR-797 isothiocyanate. The probes were sterile filtered and stored at −80 °C.

**In vivo imaging.** BALB/c nude or NOD SCID mice bearing 100 mm$^3$ subcutaneous MDA-MB-468, A431 or HepG2 xenografts were intravenously injected with PEG engager$^{CD19}$ or PEG engager$^{EGFR}$ (6 mg kg$^{-1}$), respectively. Five hours after PEG engager injection, the mice were intravenously administrated with 4arm-PEG$_{10K}$- NIR-797 probes (5 mg kg$^{-1}$). Pentobarbital anaesthetized mice were imaged with an IVIS Spectrum imaging system (excitation, 745 nm; emission, 840 nm; PerkineElmer) at 24, 48 and 72 h after injection.

**Pre-docked PEG engagers.** Characterization, in vivo blood half-life and in vivo therapy of pre-docked PEG engagers is described in Supplementary Methods.

**In vivo antitumour therapy.** Groups of NSG or NOD SCID mice bearing 44.7 ± 10.7 mm$^3$ MDA-MB-231 (n = 6) or 84.3 ± 4.3 mm$^3$ subcutaneous MDA-MB-468 (n = 8) tumours on their right flank were intravenously injected with PBS or 6 or 18 mg kg$^{-1}$ PEG engagers. After 5 h, the mice were intravenously administrated with free doxorubicin (3 mg kg$^{-1}$) or Doxisome (1 or 3 mg kg$^{-1}$). Treatment was repeated once a week for a total of 4 weeks. Tumour sizes were measured every 7 days. Tumour volumes were calculated according to the formula: length × width × height × 0.5.

**Statistical analysis.** Results are presented as the mean ± s.d. All experiments were repeated at least two times with representative data shown. Animal sample size was chosen based on similar well-characterized literature. Statistical analyses were examined using the two-way analysis of variance. Differences in tumour volumes between groups were examined for statistical significance using one-way analysis of variance followed by Dunnett's multiple comparisons; a probability value < 0.05 was considered statistically significant. No statistical method was used to predetermine sample sizes.

**Data availability.** Data supporting the findings of this study are available in the article and its Supplementary Information files, or from the corresponding author on reasonable request.

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

## Acknowledgements

This work was supported by intramural funding from Academia Sinica and a grant from the Academia Sinica Research Program on Nanoscience and Nanotechnology. We appreciate Dr Mi-Hua Tao and Dr Cheng-Pu Sun of the Institute of the Biomedical Sciences at Academia Sinica, Taipei, Taiwan for technical support in processing human hepatocytes. We thank Dr Han-Chung Wu of the Institute of Cellular and Organismic Biology at Academia Sinica, Taipei, Taiwan for providing PEGylated liposomal vinorelbine. The authors thank Dr Shu-Chuan Jao of the Biophysics Core Facility, Scientific Instrument Center at Academia Sinica, Taipei, Taiwan for assistance in performing Microscale thermophoresis and Nano DSC III experiments. We thank Ms. Show-Rong Ma of the Confocal Microscopy Core Facility, Scientific Instrument Center at the Institute of Biomedical Sciences, Academia Sinica, Taipei, Taiwan for technical support with the Axiovert 200M Confocal Microscope. We appreciate help with lentivirus production by the National RNAi Core Facility, Institute of Molecular Biology/Genomic Research Center, at Academia Sinica, Taipei, Taiwan. We also thank the Taiwan Liposome Company, Taipei, Taiwan for providing Doxisome.

## Author contributions

Y.C.S., T.L.C. and S.R.R. conceived the project and designed the experiments. Y.C.S. and P.A.B. performed experiments. P.A.B. prepared liposomes. B.M.C. and K.H.C. produced anti-PEG hybridomas. Y.C.S. and K.H.C. cloned humanized antibody genes. Y.C.S., T.L.C. and S.R.R. analysed data. Y.C.S. and S.R.R. wrote the manuscript.

## Additional information

**Competing interests:** The authors declare no competing financial interests.

