## [Peer Review File · Nature Communications]

Reviewers' comments:

Reviewer #1 (Remarks to the Author):

The authors reported a PEG engager that binds polyethylene glycol and EGFR to deliver PEGylated nanomedicines to EGFR+ TNBC cells and enhanced the antitumor activity of doxorubicin in TNBC xenograft models. Their data showed the enhanced delivery of PEGylated nanomedicines in mice.

1. What is the rationale of utilizing this strategy in TNBC? Is it unique in TNBC? Does it also enhance the delivery of PEGylated nanomedicines in other EGFR+ breast cancer subtypes?
2. It is not clear why they chose to bind EGFR on PEG engager? Is it because of the endocytosis of EGFR? The authors need more clear data to support their hypothesis that the trafficking mechanism of EGFR enhances the delivery of PEGylated nanomedicines. Then how is EGFR unique to other cell surface receptors?
3. PEG engager has a single chain of Necitumumab, an anti-human EGFR antibody. The biological effects of PEG engagerEGFR binding to EGFR on cancer cells have not been examined. It is not clear whether it inhibits EGFR signaling and whether it contributes to the reduction of tumor growth in xenografts.
4. Including doxorubicin, have they ever tested other drugs?
5. Xenograft study did not show any evidence of doxorubicin's effect.
6. Some experiments are missing proper control. For example, page 7 lines 125-136, in order to conclude that "PEG engagerEGFR can deliver PEGylated nanoparticles into TNBC cells that express EGFR", the authors should compare MDA-MB-468 cells with other cancer cells which do not have or have low EGFR expression. Also more than 1 cell line with endogenous EGFR should be examined.
7. Most experiments were conducted in 1 cell line. For example, Figs. 3, 4, and 6B should be done in at least 2 cell lines.
8. MDA-MB-468 and MDA-MB-231 xenograft models were used in Fig. 7 to demonstrate that "pre-targeting PEG engagerEGFR to EGFR over-expressing TNBC tumors can markedly enhance the therapeutic efficacy of PEGylated liposomal doxorubicin" (page 11, lines 208-211). However, MDA-MB-231 is a TNBC cell line which has extremely low expression level of EGFR and it is not a good model for their study. Further, it raises the concern whether their PEG engagerEGFR truly works in EGFR overexpressing TNBC cells.
9. Page 21, lines 367-368: PC3, SKBR3, SK-MES-1, Hut125, Caski, HT29, H2170, LS174T, HepG2, SW480, Ramos were mentioned in method section. But didn't see any experiments performed in these cell lines. Not sure if Figs. 8b and c used these cell lines. If so, need to be described.

Reviewer #2 (Remarks to the Author):

In their manuscript, Su and colleagues report a PEG engager - nanocarrier approach that may deliver doxorubicin to EGFR-expressing TNBC cells with specificity through conditional endocytosis. This is an interesting treatment approach aimed at improving specific delivery of toxic drugs like doxorubicin to enhance tumor cell killing and reduce toxic side effects of such drugs. The findings will be interesting to other experts in the field and require consideration. The manuscript requires further work to strengthen the findings and improve the quality of the data:

- 1) Introduction: lines 84-88 describe the design approach for the PEGylated liposomal doxorubicin. These lines really belong in the Results and should constitute the first Results section, please move to the Results. The final sentences of the Introduction will then need to be rephrased.
- 2) Results, lines 111-112 and Figure 2b, where the MW size of the main bands is stated. In essence, it is impossible to confirm the size of these bands since the Coomassie blue gel is overloaded. There is also a second band at a lower molecular weight which is not explained or discussed anywhere. These experiments will need to be repeated and also any bands that do not correspond to the appropriate molecular weight expected for the construct should be explained.
- 3) Results, lines 129-133 and Figure 3b:

a. How was the percent uptake calculated (out of what total)? How were the controls set up? This is unclear here and in other sections of the manuscript.

b. Important controls are missing, such as conditions with and without engager and with and without PEG-Qdot? These need to be shown.

4) Results, section on conditional internalization, lines 138-150 and Figure 4:

a. Where is the description of Figure 4a?

b. Please provide larger images of individual cells in order for the reader to distinguish between cell surface and intracellular binding, this should be much clearer by confocal microscopy, however at present, it is not possible to distinguish in the images provided.

c. What intracellular compartments do these molecules reach? A co-stain for endosomes or lysosome markers should confirm intracellular localization and also provide an indication as to where in the cells the complexes are sequestered.

5) Results, section on in vivo anti-tumor activity of the complex and Figure 7: these experiments are missing controls with PEG engager alone; what are the effects? Presumably PEG engagers localise to the tumours but do not restrict the growth of tumours?

6) Results, lines 220-223 and Figure 8c: the key experiment here is to show lack of toxicity to hepatocytes at the same and lower doses that are shown to be toxic to tumor cells.

7) What is the rationale for using subcutaneous xenograft tumor models in NSG mice? There are well-published protocols for MDA-MB-231 orthotopic models of TNBC widely described in the literature.

8) Please describe the full strain name of NSG mice.

9) Please correct typo on line 59 to "therefore".

Reviewer #3 (Remarks to the Author):

MS NCOMMS-16-14229 entitled "Conditional internalization of PEGylated nanomedicines by PEG engagers for triple negative breast cancer therapy" by Su et al. designed a protein construct which consists of an anti-PEG Fab fused to an anti-EGFR scFv. The constructs named as PEG engagers can bind to EGFR overexpression cancer cells through anti-EGFR scFv portion of the molecule, and anti-PEG Fab portion of the molecule can bring PEGylated liposomal doxorubicin nanoparticles in proximity to the cancer cell, and the cytotoxic doxorubicin is then delivered inside the cancer cell through receptor mediated endocytosis. Since PEGylation is commonly used for nanocarriers, it is plausible that the PEG engagers are applicable to any oncogenic receptors on the surface of cancer cells. The manuscript is well written and results were logically presented. I have some major and minor questions/comments on the manuscript.

1) The T_{1/2} of the PEG engager was only 2.1 hours in mice. The extreme short half-life of the constructs raises the question of its effectiveness in human disease settings. It is suggested that the authors test the PK and efficacy of a drug formulation of combining the doxisome and the PEG engager. It is possible that by docking the engagers on the nanoparticles can increase the T_{1/2} of the engagers and, as a result, enhance its efficacy.

2) The work described in the manuscript is more a drug delivery method than a treatment modality for triple negative breast cancer. It is suggested to validate the platform in additional receptor/cancer systems, such as IGF1R and HER2, to demonstrate its general utility.

3) The efficacy differential between doxisome and doxisome/PEG engager treatments was only 2-fold in both MDA-MB-468 and MDA-MB-231 tumor models. Suggest carrying out a dose titration experiment for doxisome in the in vitro and in vivo models. The titration experiments will help to answer the question whether the PEG engager can reduce the concentration of doxisome in the treatment regimens.

4) A naked doxorubicin control needs to be included in Figure 5a-c.

5) A PEG engager control was missing in Figure 7a (MDA-MB-468).

6) A naked doxorubicin control needs to be included in Figure 7a-c.

Reviewer #4 (Remarks to the Author):

Roffler and co-workers have demonstrated the development and efficacy of a recombinant PEG engagerEGFR to specifically target PEGylated nanotherapeutics to EGFR-overexpressing TNBC cells. The authors convincingly show that the PEG engagerEGFR has high binding affinity to both PEG and EGFR, and that it efficiently binds to the TNBC cell surface without internalization until PEG is recruited and binds to the engager, with a preference for the high-density EGFR TNBC cells. For both in vitro and in vivo studies, the authors show an improvement in the anti-proliferative activity of PEGylated liposomal doxorubicin (Doxisome) when pre-targeted with PEG engagerEGFR compared to the Doxisome alone. The improved efficacy observed for NSG mice bearing MDA-MB-468 or MDA-MB-231 tumors (as demonstrated by a reduction in tumor volume) was accompanied by limited side effects (indicated by no change in body weight). Appropriate negative controls were used throughout the study to show that the binding and efficacy of the PEG engagerEGFR was specific to the EGFR on TNBC cell surfaces.

The novelty of this work will be of great interest to pharmacologists, drug developers and clinicians alike. The significance of this work to the field includes the ability to implement existing, FDA-approved PEGylated therapeutics while potentially increasing efficacy at lower doses, which subsequently reduces side effects (e.g., HFS) and overall cost of treatment. Many investigators are developing approaches that functionalize the therapeutic in order to target the cancer cell, but are confronted with problems such as complexity of development and analytics, controlling the location and number of payload, specificity to the target, and reduction in 'stealth-ness'. The work presented by Roffler and colleagues largely bypasses these issues. Furthermore, a single PEGylated therapeutic can be utilized for heterogeneous tumors by incorporating engagers with specificities to other over-expressed proteins (other than EGFR) on cancer cell surfaces. Internalization of the engager-PEG complex correlates with the density of the overexpressed protein, thereby targeting the therapy to the cancer cells while potentially reducing effects on healthy cells. For these reasons, this paper will be a significant contribution to the field of anti-cancer drug development.

The methodology used to perform this study was appropriate, thorough, and utilized the appropriate techniques and controls. The data were presented in an order that created a natural flow. The data support the affinity of the engagers for their ligands, specificity to cell surface receptors, internalization correlating with the presence of PEG, enhancement of anti-tumor activity, tumor binding in vivo, and that efficacy is not compromised by pre-existing anti-PEG antibodies. The EGFR+ TNBC cell line MDA-MB-468 was used for most studies while other TNBC cell lines were used to study anti-proliferative activity of PEG therapeutics (MDA-MB-231 and BT20) and tumor size in in vivo studies (MDA-MB-231). Significant differences among data were evaluated appropriately using Student's t-test or by ANOVA with Dunnett's test. The conclusions drawn from the data are valid and well-supported by the data generated in the study. Proper credit has been given to previous work (to the best of my knowledge, along with my own literature searches).

The following includes a few questions/ comments that should be addressed by the authors to improve the paper. While I believe that the manuscript should be accepted without performing additional experiments, I have also provided suggestions for experiments to further characterize the efficacy and disposition of the PEG-engagers.

- 1) How was the 150 µg dose of engager chosen? Why was the Doxisome normalized to mouse weight but the engagers dosed at a uniform amount to all mice? Has a maximum-tolerated dose been established for eth engagers? If so, what limits the maximum dose?
- 2) The recombinant PEG engagers appear larger than the expected MW of 85 kDa. Why is this?
- 3) Would the Alexa Fluor label affect binding of the engagers to their ligands?
- 4) The PK results for both PEG engagerEGFR and PEG engagerCD19 show that approximately 10% of the engager is present in plasma at 5 hours and detectable levels are observed in the plasma at 24 hours. If the PEGylated liposomal doxorubicin is injected at 5 hours, it is likely that the engager + Doxisome will bind and circulate in the blood. Are there any toxic effects expected for this circulating engager + NP? How is the PK of the engager and/or Doxisome affected by this

interaction in the blood?

5) Several figures refer to an n=3, but error bars are not included in the plots (or they are small enough so that I cannot see them). Please check error bars on the following figures

- Figure 2d (CD19)
- Figures 5a-c
- Figure 6a
- Supplementary figure 2a

6) The authors state in lines 249-251 that '...antibodies that recognize the extracellular domain of the EGFR can target both wild-type and mutated EGFRs.' Has the binding of engager, and, subsequently, the PEG therapeutic, been demonstrated for models that bear activating EGFR mutations in TNBC?

7) It would be very interesting to evaluate the disposition of the engagers in tissue - has this been evaluated? Furthermore, it is well understood that the MPS system is involved in the clearance of nanoparticles and biotherapeutics. Have the authors evaluated the interaction of the engagers with the MPS? This would be valuable information in understanding the PK of the engagers.

8) Would the authors comment on the presence of 4armPEG10k-NIR-797 in the MDA-MB-468 xenograft tumors in NSG mice that were dosed with PEG engagerCD19 (negative control; see Figure 6b). Is this attributed to EPR of the PEG structure, or is it due to binding of the PEG engagerCD19 to the tumor? Furthermore, why is there fluorescence observed in the head/neck region for 3 of 6 mice receiving PEG engagerEGFR?

9) Treatment of TNBC brain metastases is a topic of current research for PEGylated liposomal doxorubicin. Have the authors performed studies in intracranial models? Is there any evidence of how the engagers affect drug accumulation within the blood brain barrier?

In summary, the manuscript was well-written with an orderly flow. The appropriate experiments were performed to draw the conclusions made by the authors. The data, discussion and conclusions were presented so that they could be understood by a diverse audience. I recommend the manuscript for publication and look forward to following the development of these novel PEG engagers.

Reviewer #1 (Remarks to the Author):

The authors reported a PEG engager that binds polyethylene glycol and EGFR to deliver PEGylated nanomedicines to EGFR⁺ TNBC cells and enhanced the antitumor activity of doxorubicin in TNBC xenograft models. Their data showed the enhanced delivery of PEGylated nanomedicines in mice.

1. What is the rationale of utilizing this strategy in TNBC? Is it unique in TNBC? Does it also enhance the delivery of PEGylated nanomedicines in other EGFR⁺ breast cancer subtypes?

We examined TNBC due to the need for improved treatments for this type of cancer. However, we believe that this approach can be generalized to other tumor types and antigen targets. To address this issue, we further examined the anti-proliferation activity of PEG-liposomal doxorubicin in SKBR3 (EGFR⁺, non-TNBC breast cancer cells) and PC9 (EGFR⁺, lung cancer cells with mutant EGFR, exon19del E746–A750) cells. Our results show that the PEG engager can also enhance the anti-proliferative activity of PEG-liposomal doxorubicin to non-TNBC EGFR⁺ cancer cells.

We added the related description in Materials and method (Cell proliferation assay) on page 32 line 585.

“PC3, SKBR3, SK-MES-1, Hut125, Caski, HT29, H2170, LS174T, HepG2, SW480 MDA-MB-231, MDA-MB-468, BT-20 and PC9 cells (10,000 cells/well) were seeded in 96-well plates overnight. **Serial dilutions of free doxorubicin or vinorelbine were directly added to the cells as positive controls.**”

We also modified the description in Results section (PEG engager-directed liposomal anti-cancer drugs can effectively inhibit proliferation of EGFR-positive cancer cells) on page 9 line 156.

“We next investigated whether PEG engager^{EGFR} can enhance the anti-proliferation activity of a drug-loaded nanocarrier in EGFR-positive cancer cells. MDA-MB-231, MDA-MB-468 and BT-20 (EGFR⁺, TNBC), SKBR3 (EGFR⁺, non-TNBC breast cancer) and PC9 (EGFR⁺, non-small cell lung cancer) cells were incubated with PEG engager^{EGFR} or PEG engager^{CD19} as a negative control and subsequently treated with graded concentrations of free drug (doxorubicin or vinorelbine), empty liposomes, Doxisome (PEG-liposomal doxorubicin) or Lipo-Vino (PEG-liposomal vinorelbine). PEG engager^{EGFR} significantly enhanced the anti-proliferation activity of Doxisome (Fig. 5 and Supplementary Fig. 4) and Lipo-Vino (Supplementary Fig. 5) against EGFR-positive cancer cells as compared with drug-loaded nanocarrier alone, drug-loaded nanocarrier plus PEG engager^{CD19} or empty liposomes with PEG engager^{EGFR}.”

2. It is not clear why they chose to bind EGFR on PEG engager? Is it because of the endocytosis of EGFR? The authors need more clear data to support their hypothesis that the trafficking mechanism of EGFR enhances the delivery of PEGylated nanomedicines. Then how is EGFR unique to other cell surface receptors?

We choose this target for two main reasons. First, EGFR is expressed on about 50% of TNBC cells, and may therefore be a good target to treat this type of cancer. Second, we previously demonstrated that EGFR may be a good target for nanomedicine therapy (*ACS nano* 10(1), 648-662 (2016)). In that study, we engineered chimeric anti-PEG receptors to mimic endocytosis of PEG-nanoparticles (PEG-NPs) targeted to HER1 (EGFR) or HER2. Although both anti-PEG receptors possess similar PEG-binding activity, cells that express anti-PEG-HER1 receptors showing greater uptake and higher cytotoxicity of Doxisome as compared to the cells that express anti-PEG-HER2 receptors. These results suggest that EGFR may be a good candidate to target liposomal drugs for cancer therapy. However, EGFR is not unique in the sense that other surface receptors may be appropriate targets for other types of cancer.

We also performed an experiment using EGFR specific shRNA to knockdown EGFR expression in BT20 cells. We found that the anti-proliferation activity of PEGylated medicines (Doxisome) is significantly decreased in EGFR-knockdown BT20 cells (Supplementary Fig. 6) as compared to parental BT20 cells (Fig. 5a), indicating that EGFR is required for the anticancer activity of PEG engager^{EGFR} targeted nanoparticles.

We added a description of the experimental details in Supplementary Methods (shRNA transfection).

“The small hairpin RNA (shRNA) plasmid for the EGFR gene was obtained from the National RNAi Core Facility (Academia Sinica, Taipei, Taiwan). For EGFR knockdown, BT-20 cells were seeded overnight in 6-well plates at a density of 1×10^5 cells per well. Fresh medium without serum or antibiotics containing 5 $\mu\text{g/mL}$ Polybrene (Sigma-Aldrich) and lentivirus carrying shRNA targeting EGFR (multiplicity of infection = 10, prepared by the National RNAi Core Facility) was added to the cells for 24 h. After lentiviral infection, the cells were selected in 2 $\mu\text{g/mL}$ puromycin for 4 days.”

We modified the description in the Results section (PEG engager-directed liposomal anti-cancer drugs can effectively inhibit proliferation of EGFR-positive cancer cells) on page 10 line 172.

“In contrast to wild-type BT20 cells, PEG engager^{EGFR} did not enhance the anti-proliferation activity of Doxisome in BT20/shEGFR cancer cells (BT20 cells treated with shRNA to knock down the expression of EGFR) as compared with drug-loaded nanocarrier alone or drug-loaded nanocarrier plus PEG engager^{CD19} (Supplementary Fig. 6), further showing that PEG engager^{EGFR} mediates endocytosis of PEGylated nanocarriers via the EGFR internalization pathway.”

We also performed additional live cell imaging studies to show that the PEG engager^{EGFR} could selectively enhance the endocytosis of PEGylated nanoparticles into cancer cells that express EGFR (Fig. 3a and 3b) but not in EGFR⁻ cells (Fig. 3c). We added the related description in Materials and method (Confocal microscopy of PEG engager-targeted PEGylated nanoparticles) on page 29 line 535.

“The coverslips were washed twice with PBS and then 5×10^4 MDA-MB-468 (EGFR⁺), A431 (EGFR⁺), BT-20 (EGFR⁺) or MCF-7 (EGFR⁻) cancer cells were seeded on the coverslips.”

We also modified the description in the Results section (PEG engagers can selectively deliver PEGylated nanoparticles to antigen-positive cancer cells) on page 7.

“Cancer cell-specific uptake of PEGylated nanoparticles mediated by PEG engagers was examined by real-time confocal microscopy cell imaging of **EGFR-negative MCF7 cells or EGFR-positive MDA-MB-468 and A431 cells** treated stepwise with PEG engager^{EGFR} or PEG engager^{CD19} and then fluorescent PEG-Qdot655. **Both MDA-MB-468 TNBC and A431 cells express EGFR but not CD19. MCF7 cells express neither EGFR or CD19.** PEG engager^{EGFR} mediated rapid accumulation of PEG-Qdot655 in **both MDA-MB-468 and A431 cells** (Fig. 3a and 3b, upper panels), **but not in MCF7 cells** (Fig. 3c, upper panels). By contrast, no uptake of PEG-Qdot655 was observed in MDA-MB-468, **A431 and MCF7 cells** treated with **control** PEG engager^{CD19} (Fig. 3, lower panels). We conclude that PEG engager^{EGFR} can deliver PEGylated nanoparticles into TNBC cells that express EGFR.”

3. PEG engager has a single chain of Necitumumab, an anti-human EGFR antibody. The biological effects of PEG engager^{EGFR} binding to EGFR on cancer cells have not been examined. It is not clear whether it inhibits EGFR signaling and whether it contributes to the reduction of tumor growth in xenografts.

This is a very interesting question. To address this issue, EGF-stimulated EGFR-positive A431 cancer cells were incubated with or without PEG engagers, Herceptin (anti-HER2 IgG) or Erbitux (anti-EGFR IgG). We found that both Erbitux and PEG engager^{EGFR} inhibited the phosphorylation of EGFR and Erk as compared to the negative controls Herceptin and PEG engager^{CD19} (Supplementary Fig. 10).

We added a description of the experimental details in Supplementary Methods (Western blot analysis).

“**A431 cells were starved in DMEM without serum for 18 h. The cells (2×10⁵ cells/group) were detached with Accutase (Innovative Cell Technologies) and incubated with or without 50 nM of Herceptin (anti-HER2, Genentech), Erbitux (anti-EGFR, Merck), PEG engager^{CD19} or PEG engager^{EGFR} prepared in PBS at 37 °C for 30 min prior to stimulation with or without recombinant human EGF (5 nM, R&D Systems) at 37 °C for 5 min. The cells were lysed by using Pierce™ IP Lysis Buffer (ThermoFisher Scientific) containing Halt™ Protease and Phosphatase Inhibitor Cocktail (ThermoFisher Scientific). The protein concentration was analyzed using a BCA protein assay (ThermoFisher Scientific). Forty micrograms of total proteins were electrophoresed on a SDS-PAGE gel, transferred to a nitrocellulose membrane, and probed with anti-EGFR (Santa Cruz Biotechnology, catalogue # SC-120), anti-phospho EGFR (Tyr1068) (Cell Signaling Technology, catalogue #2236),**

anti-phospho Erk (Cell Signaling Technology, catalogue #9101), or anti-alpha tubulin (ThermoFisher Scientific, catalogue # RB-9281-P1) antibodies.”

A description of this experiment was added to the Results section (Anti-tumor activity of pre-targeted PEG engager^{EGFR} combined with Doxisome) on page 12 line 218.

“To investigate whether PEG engager^{EGFR} can inhibit EGFR signaling, EGFR-positive A431 cells were stimulated with or without EGF and then co-incubated with PEG engagers or control antibodies. Both Erbitux (monoclonal anti-EGFR IgG) and PEG engager^{EGFR} inhibited the phosphorylation of EGFR and Erk as compared to negative control Herceptin (anti-HER2 IgG) and PEG engager^{CD19} (Supplementary Fig. 10).”

Investigation of direct anti-cancer activity by the PEG engager^{EGFR} *in vivo* showed that it did not produce significant antitumor activity as compared to treatment with PBS alone (open diamonds in Figs. 7a and 7c).

We modified the description in Material and methods section on page 35 line 640.

“Groups of NSG or NOD SCID mice bearing $44.7 \pm 10.7 \text{ mm}^3$ MDA-MB-231 (n=6) or $84.3 \pm 4.3 \text{ mm}^3$ subcutaneous MDA-MB-468 (n=8) tumors on their right flank were intravenously injected with PBS or 6 mg/kg or 18 mg/kg PEG engagers. After five hours, the mice were i.v. administrated with 1 mg/kg or 3 mg/kg Doxisome. Treatment was repeated once a week for a total of 4 weeks.”

We also modified Results (Anti-tumor activity of pre-targeted PEG engager^{EGFR} combined with Doxisome) to include this result on page 13 line 227.

“... Although engager^{EGFR} inhibited EGFR signaling *in vitro*, mice treated only with PEG engager^{EGFR} displayed similar tumor growth as mice treated with PBS (Figs. 7a and 7c) ...”

4. Including doxorubicin, have they ever tested other drugs?

To address this issue, we also examined the anti-proliferation activity of PEG-liposomal vinorelbine in EGFR-positive TNBC cells. The results show that PEG engager can enhance the anti-proliferative activity of PEG-liposomal vinorelbine to TNBC EGFR⁺ cancer cells (Supplementary Fig. 5).

We added the related description (Cell proliferation assay) to include this experimental detail on page 32 line 589.

“... PEG engager^{EGFR} antibodies were added to the cells for 30 min at 37°C followed by addition of graded concentrations of PEGylated liposomal doxorubicin (Doxisome®, 13.9 μmol/mL lipid concentration, Taiwan Liposome Company Ltd., Taipei, Taiwan) or liposomal vinorelbine⁷² (kindly provide by Dr. Han-Chung Wu, Research Fellow, Institute of Cellular and Organismic Biology, Academia Sinica, Taiwan.) ...“

We also modified Results (PEG engager-directed liposomal doxorubicin anti-cancer drugs can effectively inhibit proliferation of TNBC EGFR-positive cancer cells) to include this result on page 9 line 163.

“... PEG engager^{EGFR} significantly enhanced the anti-proliferation activity of Doxisome (Fig. 5 and Supplementary Fig. 4) and Lipo-Vino (Supplementary Fig. 5) against EGFR-positive cancer cells as compared with drug-loaded nanocarrier alone, drug-loaded nanocarrier plus PEG engager^{CD19} or empty liposomes with PEG engager^{EGFR} ...“

5. Xenograft study did not show any evidence of doxorubicin's effect.

We repeated the in vivo experiments including free doxorubicin treatment in xenograft studies in the updated Figure 7. We found that a high dose of free doxorubicin (3 mg/kg) slightly inhibited tumor growth as compared to PBS control (green triangles in Fig. 7a). However, a low dose of liposomal doxorubicin alone (1 mg/kg) showed better anti-tumor effects than free doxorubicin treatment (red squares in Fig. 7a).

We added the modified description in Materials and methods (In vivo antitumor therapy) on page 35 (line 643).

“... After five hours, the mice were i.v. administrated with free doxorubicin (3 mg/kg)

or Doxisome (1 mg/kg or 3 mg/kg).

A description of this experiment was added to the Results section (Anti-tumor activity of pre-targeted PEG engager^{EGFR} combined with Doxisome) on page 13 line 229.

“... Free doxorubicin suppressed tumor growth as compared to treatment of mice with PBS (Fig. 7a, $p < 0.05$). PEG engager^{CD19} combined with 1 mg/kg Doxisome or 1 mg/kg Doxisome alone displayed similar and better suppression of tumor growth as compared to treatment of mice with free doxorubicin or PBS vehicle (Fig. 7a, $p < 0.0001$ and 7c, $p < 0.001$).”

6. Some experiments are missing proper control. For example, page 7 lines 125-136, in order to conclude that "PEG engager^{EGFR} can deliver PEGylated nanoparticles into TNBC cells that express EGFR", the authors should compare MDA-MB-468 cells with other cancer cells which do not have or have low EGFR expression. Also more than 1 cell line with endogenous EGFR should be examined.

Thank you for this suggestion. To address this issue, we included EGFR-negative MCF7 and EGFR-positive A431 cells. The results show that PEG engager^{EGFR} can specifically deliver PEGylated nanoparticles into MDA-MB-468 TNBC and A431 cells that express EGFR (Fig. 3a and 3b) but not in EGFR-negative MCF7 cells (Fig. 3c)

We modified the description in the Results section (PEG engagers can selectively deliver PEGylated nanoparticles to antigen-positive cancer cells) at page 7 line 123.

“Cancer cell-specific uptake of PEGylated nanoparticles mediated by PEG engagers was examined by real-time confocal microscopy cell imaging of EGFR-negative MCF7 cells or EGFR-positive MDA-MB-468 and A431 cells treated stepwise with PEG engager^{EGFR} or PEG engager^{CD19} and then fluorescent PEG-Qdot655. Both MDA-MB-468 TNBC and A431 cells express EGFR but not CD19. MCF7 cells express neither EGFR or CD19. PEG engager^{EGFR} mediated rapid accumulation of PEG-Qdot655 in both MDA-MB-468 and A431 cells (Fig. 3a and 3b, upper panels), but not in MCF7 cells (Fig. 3c, upper panels). By contrast, no uptake of PEG-Qdot655 was observed in MDA-MB-468, A431 and MCF7 cells treated with control PEG engager^{CD19} (Fig. 3, lower panels). We conclude that PEG engager^{EGFR} can deliver PEGylated nanoparticles into TNBC cells that express EGFR. “

We modified the related description in Materials and Methods section (Confocal microscopy of PEG engager-targeted PEGylated nanoparticles) at page 29 line 535.

“The coverslips were washed twice with PBS and then 5×10^4 MDA-MB-468 (EGFR⁺), A431 (EGFR⁺), BT-20 (EGFR⁺) or MCF-7 (EGFR⁺) cancer cells were seeded on the coverslips.”

7. Most experiments were conducted in 1 cell line. For example, Figs. 3, 4, and 6B should be done in at least 2 cell lines.

We included two EGFR-positive cells (MBA-MBD-468 and A431) to perform real-time confocal cell imaging experiments in Figure 3. We found that PEG engager^{EGFR} can specifically deliver PEGylated nanoparticles into both MDA-MB-468 (Fig. 3a) and A431 (Fig. 3b) cells but not the control PEG engager^{CD19} (Fig. 3 lower panels).

We also included EGFR-positive MDA-MB-468 and BT20 TNBC cell groups in real-time confocal cell imaging experiments in Fig. 4, supplementary Fig. 2 and supplementary Fig. 3. These results show that the binding of PEG engager^{EGFR} on MDA-MB-468 (Fig. 4a, Supplementary Fig. 3a and 3b) and BT-20 cells (Supplementary Fig. 2a) revealed limited internalization after 1 h or 9 h incubation time. However, subsequently added PEGylated nanoparticles (PEG-Qdot655) could be bound by PEG engager^{EGFR} on the cell membrane and then rapidly internalized (Fig. 4b, Supplementary Fig. 2b and Supplementary Fig. 3b).

We modified the description in the Results section (PEG engager induces conditional internalization of PEGylated nanoparticles in EGFR⁺ TNBC cells) on page 8 line 140.

“Alexa Fluor 647-labeled PEG engager^{EGFR} was first added to live cancer cells for 1 h. Confocal imaging revealed that the engager remained on the plasma membrane of MDA-MB-468 (Fig. 4a and Supplementary Fig. 3a) and BT-20 (Supplementary Fig. 2a) cells at 37°C for 1 h with almost no internalization (Alexa Fluor 647 appears in green pseudo color). PEG engager^{EGFR} displayed limited endocytosis in MDA-MB-486 cells even after 9 h (Supplementary Fig. 3b). PEG-Qdot655 added to the cells, however, bound to PEG engager^{EGFR} on the cell

membrane and were rapidly internalized into the cells (Fig. 4b, Supplementary Fig. 2b and Supplementary Fig. 3b, red color).”

A description of the experimental details was added in the Materials and methods section (Confocal microscopy of PEG engager-targeted PEGylated nanoparticles) on page 30 line 545.

“Similarly, conditional internalization of PEGylated nanoparticles was determined by incubating MDA-MB-468 or BT-20 cells with 10 µg/mL of Alexa Fluor 647 conjugated PEG engager^{EGFR} (excitation/ emission, 650 nm/ 675 nm) at 37°C for 30 min in medium ...”

Finally, we included two EGFR^{High} xenograft models (MDA-MB-468 and A431) and one EGFR^{Low} (HepG2) xenograft model in IVIS imaging experiments in Figure 6. The results show that PEG engager^{EGFR} enhances the uptake of 4armPEG_{10k}-NIR-797 probes in both MDA-MB-468 (Fig. 6) and A431 xenografts (Supplementary Fig. 8) as compared to HepG2 xenografts (Supplementary Fig. 9).

We modified the description in the Results section (Pharmacokinetics and tumor targeting of PEG engager) at page 11 line 204.

“To visualize whether pre-targeting can facilitate the uptake and retention of PEGylated compounds in tumors, mice bearing established EGFR^{High} (MDA-MB-468 and A431) or EGFR^{Low} (HepG2) tumors were i.v. injected with 6 mg/kg PEG engager and then subsequently i.v. injected with 4armPEG_{10k}-NIR-797 probe 5 h later. IVIS optical imaging of these mice at 24, 48, and 72 h after probe injection showed that the fluorescence signal in PEG engager^{EGFR} targeted tumors was significantly enhanced as compared to the PEG engager^{CD19} control group (Fig. 6b and Supplementary Fig. 8). The fluorescent intensity in PEG engager^{EGFR} targeted tumors at 24, 48 and 72 h was 2.7-fold, 2.1-fold and 2.8-fold greater than in the control PEG engager^{CD19} treated tumors, respectively (Fig. 6c). Neither PEG engager^{EGFR} nor PEG engager^{CD19} enhanced the fluorescence signal in HepG2 (EGFR^{Low}) tumor-bearing mice (Supplementary Fig. 9).”

We modified the description in Materials and methods section (IVIS imaging) at page 34 line 631.

“BALB/c nude or NOD SCID mice bearing 100 mm³ subcutaneous MDA-MB-468, A431 or HepG2 xenografts were intravenously (i.v.) injected with

PEG engager^{CD19} or PEG engager^{EGFR} (6 mg/kg), respectively.”

8. MDA-MB-468 and MDA-MB-231 xenograft models were used in Fig. 7 to demonstrate that "pre-targeting PEG engager^{EGFR} to EGFR over-expressing TNBC tumors can markedly enhance the therapeutic efficacy of PEGylated liposomal doxorubicin" (page 11, lines 208-211). However, MDA-MB-231 is a TNBC cell line which has extremely low expression level of EGFR and it is not a good model for their study. Further, it raises the concern whether their PEG engager^{EGFR} truly works in EGFR overexpressing TNBC cells.

We have determined that our MDA-MB-231 cells express similar EGFR levels to MDA-MB-468 cells as shown below. Other studies also reported that MDA-MB-231 cells express EGFR (Oncogene. 2011, 30, 770–780; Br J Cancer. 2004, 91, 795-802).

9. Page 21, lines 367-368: PC3, SKBR3, SK-MES-1, Hut125, Caski, HT29, H2170, LS174T, HepG2, SW480, Ramos were mentioned in method section. But didn't see any experiments performed in these cell lines. Not sure if Figs. 8b and c used these cell lines. If so, need to be described.

We appreciate the reviewer for pointing out the issue. These cell lines were used for the experiment shown in Figure 8. We added the description in the Materials and methods section (Flow cytometer analysis) at page 29. We actually did not use Ramos cells so we also deleted Ramos.

Reviewer #2 (Remarks to the Author):

In their manuscript, Su and colleagues report a PEG engager - nanocarrier approach that may deliver doxorubicin to EGFR-expressing TNBC cells with specificity through conditional endocytosis. This is an interesting treatment approach aimed at improving specific delivery of toxic drugs like doxorubicin to enhance tumor cell killing and reduce toxic side effects of such drugs. The findings will be interesting to other experts in the field and require consideration. The manuscript requires further work to strengthen the findings and improve the quality of the data:

1) Introduction: lines 84-88 describe the design approach for the PEGylated liposomal doxorubicin. These lines really belong in the Results and should constitute the first Results section, please move to the Results. The final sentences of the Introduction will then need to be rephrased.

We thank the reviewer for this suggestion. We moved the description from lines 84-88 to the first line of the Results section.

We also modified the final sentences of the introduction at line 84.

“This was accomplished by generating bispecific PEG-binding antibodies (PEG engagers) for targeted delivery of PEGylated nanomedicines to tumors.”

2) Results, lines 111-112 and Figure 2b, where the MW size of the main bands is stated. In essence, it is impossible to confirm the size of these bands since the Coomassie blue gel is overloaded. There is also a second band at a lower molecular weight which is not explained or discussed anywhere. These experiments will need to be repeated and also any bands that do not correspond to the appropriate molecular weight expected for the construct should be explained.

We repeated this SDS-PAGE experiments with reasonable loading amount (3 $\mu\text{g}/\text{lane}$). A therapeutic monoclonal IgG, Erbitux, was included as a positive protein control in SDS-PAGE experiments to compare molecular sizes as well as the purity of proteins. We also indicated the expected bands by using illustrations (Fig. 2b, right panel).

We noticed that the protein ladder is not very accurate under reducing conditions. Thus, the protein size of samples under non-reducing condition does not precisely match to the protein ladder. For example, Erbitux IgG runs at nearly 200 kDa in relationship referenced to the protein ladders (Fig. 2b, right panel).

Therefore, we further determined the precise molecular weight of the PEG engagers by MALDI-TOF. The PEG engager^{CD19} and PEG engager^{EGFR} have molecular weights of 78 kDa and 79 kDa, respectively (Fig. 2c).

We modified the description in Results section (Characterization of bispecific PEG engagers) at page 6 line 97 and line 105.

“Briefly, a humanized anti-PEG (6.3) Fab was constructed as a single open reading frame by fusing V_L-C_κ and V_H-CH₁ domains with an IRES bicistronic expression linker ...”

“PEG engagers purified from the culture medium display the expected molecular sizes as visualized on a reducing and a non-reducing 10% SDS-PAGE (Fig. 2b). The PEG engager^{CD19} and PEG engager^{EGFR} have molecular weights of 78 kDa and 79 kDa, respectively, as determined by MALDI-TOF (Fig. 2c).”

We added a description in the Materials and Methods section (Analysis of purified PEG engager antibodies by SDS-PAGE and MALDI-TOF) on page 25 line 486.

“Three microgram of Eributx, purified PEG engager^{CD19} or PEG engager^{EGFR} were electrophoresed in a 10% SDS-PAGE gel under reducing or non-reducing conditions and then stained by Coomassie Blue. For further analytical characterization, the antibodies were dialyzed into water and analyzed by MALDI-TOF (New ultrafleXtreme, Bruker).”

3) Results, lines 129-133 and Figure 3b:

a. How was the percent uptake calculated (out of what total)? How were the controls set up? This is unclear here and in other sections of the manuscript.

The percentages of internalized PEG engagers and PEG-Qdots were calculated by dividing the fluorescence of the intracellular regions by the whole-cell fluorescence based on the bright field cell images (page 30, line 554).

Total area: whole-cell fluorescence.

Intracellular area: Intracellular fluorescence

b. Important controls are missing, such as conditions with and without engager and with and without PEG-Qdot? These need to be shown.

Thanks for this suggestion. We treated EGFR-positive cells (MDA-MB-468 and A431) or EGFR-negative cells (MCF7) with PEG engager^{EGFR} or control PEG engager^{CD19} in real-time cell imaging in Figure 3. We found that PEG engager^{EGFR} can specifically delivery PEG-Qdot into both MDA-MB-468 (Fig. 3a) and A431 (Fig. 3b) cells but not control MCF7 cells (Fig. 3c). All cells treated with PEG engager^{CD19} failed to uptake PEG-Qdot (Fig. 3, lower panels).

In Fig. 4 (MDA-MB-468 cells) and supplementary Fig. 2 (BT20 cells), PEG engager^{EGFR} treated cells were incubated with or without PEG-Qdot. The results showed that PEG engager^{EGFR} alone could not trigger endocytosis till the addition of PEG-Qdot promoted internalization of PEG-Qdot in both MDA-MB-468 and BT20 cells.

We modified the description in the Results section on page 8 line 140 and in the Materials and methods section on page 30 line 545 as previously described.

4) Results, section on conditional internalization, lines 138-150 and Figure 4:
a. Where is the description of Figure 4a?

We re-organized Figure 4. “Original Figure 4” was changed to “supplementary Figure 3”. The description is on page 8 line 138-142.

b. Please provide larger images of individual cells in order for the reader to distinguish between cell surface and intracellular binding, this should be much clearer by confocal microscopy, however at present, it is not possible to distinguish in the images provided.

Thank you for this suggestion. We enlarged the confocal images in Figure 4.

c. What intracellular compartments do these molecules reach? A co-stain for endosomes or lysosome markers should confirm intracellular localization and also provide an indication as to where in the cells the complexes are sequestered.

This is a very useful suggestion. We repeated confocal cell imaging experiments in the presence of lysotracker staining in Figure 4. The results show that PEG

engager^{EGFR} conditionally stimulates endocytosis of PEG-Qdot and localized in lysosomes (Fig. 4c, 4d and Supplementary Fig. 3c)

We modified Results (PEG engager induces conditional internalization of PEGylated nanoparticles in EGFR⁺ TNBC cells) to include this result on page 9 line 148.

“Colocalization of PEG engager^{EGFR} and PEG-Qdot655 or lysosomes (Fig. 4b, Supplementary Fig. 2b and Supplementary Fig. 3b) verified that PEG engager^{EGFR} can conditionally stimulate endocytosis of PEGylated nanoparticles and then localize in lysosomes (Fig. 4c, 4d and Supplementary Fig. 3c).”

A description of the experimental details was added in the Materials and methods section (Confocal microscopy of PEG engager-targeted PEGylated nanoparticles) on page 30 line 549.

“... in medium containing 1 µg/mL of Hoechst 33342 and 100 nM of LysoTracker Red DND-99 (appears in purple pseudo color).”

5) Results, section on *in vivo* anti-tumor activity of the complex and Figure 7: these experiments are missing controls with PEG engager alone; what are the effects? Presumably PEG engagers localise to the tumours but do not restrict the growth of tumours?

We repeated this experiment to investigate if the PEG engager^{EGFR} by itself could produce anti-cancer activity *in vivo* in Figure 7. However, the PEG engager alone did not produce significant antitumor activity as compared to treatment with PBS alone (open diamonds in Figs. 7a and 7c).

We modified the description in the Material and methods section on page 35 line 640 and in the results section on page 13, line 227.

Although engager^{EGFR} inhibited EGFR signaling *in vitro*, mice treated only with PEG engager^{EGFR} displayed similar tumor growth as mice treated with PBS (Fig. 7a and Fig 7c).

6) Results, lines 220-223 and Figure 8c: the key experiment here is to show lack of toxicity to hepatocytes at the same and lower doses that are shown to be toxic to tumor cells.

To examine if PEG engager coupled with PEGylated medicines is toxic to cells that express low EGFR levels such as normal hepatocytes, we used HepG2 cells, which are a hepatocellular carcinoma cell line that express low levels of EGFR. We determined that PEG engagers did not enhance the anti-proliferation activity of PEGylated liposomal doxorubicin to HepG2 cells (Supplementary Fig. 4c).

7) What is the rationale for using subcutaneous xenograft tumor models in NSG mice? There are well-published protocols for MDA-MB-231 orthotopic models of TNBC widely described in the literature.

We appreciate the reviewer's suggestion. We found some protocols for MDA-MB-231 orthotopic models, however, for proof of concept we chose the relative simple subcutaneous xenograft tumor model. In the future, orthotopic models of TNBC would be a good choice.

8) Please describe the full strain name of NSG mice.

We described the full strain name of NSG, NOD SCID and BALB/c nude mice on page 24 line 432.

9) Please correct typo on line 59 to "therefore".

We appreciated reviewer to point out this issue and we corrected it. Page 3, line 59.

Reviewer #3 (Remarks to the Author):

MS NCOMMS-16-14229 entitled "Conditional internalization of PEGylated nanomedicines by PEG engagers for triple negative breast cancer therapy" by Su et al. designed a protein construct which consists of an anti-PEG Fab fused to an anti-EGFR scFv. The constructs named as PEG engagers can binds to EGFR overexpression cancer cells through anti-EGFR scFv portion of the molecule, and anti-PEG Fab portion of the molecule can bring PEGylated liposomal doxorubicin nanoparticles in proximity to the cancer cell, and the cytotoxic doxorubicin is then delivered inside the cancer cell through receptor mediated endocytosis. Since

PEGylation is commonly used for nanocarriers, it is plausible that the PEG engagers are applicable to any oncogenic receptors on the surface of cancer cells. The manuscript is well written and results were logically presented. I have some major and minor questions/comments on the manuscript.

1) The T1/2 of the PEG engager was only 2.1 hours in mice. The extreme short half-life of the constructs raises the question of its effectiveness in human disease settings. It is suggested that the authors test the PK and efficacy of a drug formulation of combining the doxisome and the PEG engager. It is possible that by docking the engagers on the nanoparticles can increase the T1/2 of the engagers and, as a result, enhance its efficacy.

The pre-targeting strategy is accomplished in a two-step process. In the first step, tumors are targeted by injecting PEG engager and waiting for excess engager to be eliminated from the blood to achieve higher tumor/non-tumor ratios. In the second step, a therapeutic compound is administered that can be captured by the pre-targeted engagers in tumors. Thus, a shorter half-life of PEG engager might be beneficial for the pre-targeting system to avoid side effects.

To investigate the effect of docking engagers directly on the nanoparticles, we first mixed different molar ratios of PEG engager and PEG-lipid on Doxisome and tested the influence on in vitro anti-proliferation activity. The results show that a molar ratio of PEG engager/PEG-lipid on Doxisome of 1:55 for pre-decoration of PEGylated Doxisome is effective (Supplementary Fig. 11a).

We then tested the pharmacokinetics of the PEG engagers on pre-docked liposomes. The results show that the serum half-life of PEG engagers using pre-docking increased (~ 3.5 h) (Supplementary Fig. 11b).

We performed an anti-tumor activity experiment using pre-docking strategy. The results show that pre-mixing of PEG engager^{EGFR} with Doxisome significantly enhanced anti-tumor efficacy (Supplementary Fig. 11c) as compared to unmodified Doxisome. We didn't observe a clear difference between pre-targeting and pre-docking approaches.

We modified the description in the Results section (Anti-tumor activity of pre-targeted PEG engager^{EGFR} combined with Doxisome) on page 14 line 243.

“We further investigated whether pre-docking of PEGylated nanoparticles with PEG engagers could enhance their therapeutic efficacy. We used a molar ratio of PEG engager and PEG-lipid on Doxisome of 1:55 (Supplementary Fig. 11a). Mice were

administrated with a mixture of PEG engager and Doxisome (pre-incubated at 4 °C for 1 h) and blood samples were then periodically collected from the tail vein of the mice. The half-lives of the PEG engagers as determined by quantitative ELISA were approximately 3.5 h (PEG engager^{EGFR}) and 3.8 h (PEG engager^{CD19}) (Supplementary Fig. 11b) after i.v. administration of PEG engager-docked Doxisome (containing 30 µg PEG engager). To examine the therapeutic activity of PEG engager-docked Doxisome, NOD SCID mice bearing human MDA-MB-468 TNBC xenografts were intravenously injected with PBS, 3 mg/kg free doxorubicin, 6 mg/kg PEG engager^{EGFR} alone, PEG engager^{CD19} decorated Doxisome or PEG engager^{EGFR} decorated Doxisome (1 mg/kg of doxorubicin) on days 1, 8, 15 and 22. Doxorubicin slightly inhibited tumor growth while significantly better anti-tumor activity was observed in mice treated with Doxisome (1 mg/kg) or PEG engager^{CD19} decorated Doxisome as compared to mice treated with PBS vehicle (Supplementary Fig. 11c, p <0.005). A higher dose (3 mg/kg) of Doxisome was toxic to the mice (Fig. 7b and Supplementary Fig. 11d). PEG engager^{EGFR} decorated Doxisome significantly suppressed TNBC tumor growth as compared to mice treated with Doxisome alone (Supplementary Fig. 11c, p < 0.05). Thus, pre-docking PEG engager^{EGFR} on Doxisome markedly enhanced therapeutic efficacy with minimal side effects as measured by body weight loss (Supplementary Fig. 11d).“

We modified the description in Supplementary Methods section (Optimized decoration of Doxisome with PEG engagers).

“Doxisome (Taiwan Liposome Company Ltd., Taipei, Taiwan) and PEG engagers were mixed at different molar ratio of protein to PEG-lipids ranging from 1: 18.3 to 1: 110 at 4 °C for 1 hour. MDA-MB-468 cells (10,000 cells/well) were seeded in 96-well plates overnight. Serial dilutions of PEG engager^{EGFR} decorated Doxisome was added to the cells in triplicate at 37°C for 4 h. The cells were subsequently washed once and incubated for an additional 72 h in fresh culture medium and then pulsed for 18 h with ³H-thymidine (1 µCi/well). Results are expressed as percent inhibition of ³H-thymidine incorporation into cellular DNA in comparison to untreated cells.“

We added the description in Supplementary Methods (In vivo pharmacokinetics).

“NOD SCID mice were intravenously injected with PEG engager^{CD19} or PEG engager^{EGFR} decorated Doxisomes (1.5 mg/kg of PEG engagers) and blood samples were periodically collected from the tail vein of the mice. Plasma was prepared by

centrifugation (5 min, $12,000 \times g$). The PEG engager levels in plasma were determined by quantitative sandwich ELISA. Maxisorp 96-well microplates were coated with 50 μL /well of anti-PEG antibody (AGP4)² (10 $\mu\text{g}/\text{mL}$) in bicarbonate buffer, pH 8.0 for 4 h at 37°C and then at 4°C overnight. The plates were blocked with 200 μL /well 5% skim milk in PBS for 2 h at room temperature and then washed with PBS three times. Serial dilutions of PEG engager decorated Doxisome (as the standards) or plasma samples in dilution buffer (2% skim milk in PBS) were added to the wells for 2 h at room temperature. After washing with PBS four times, the plates were sequentially stained with 50 μL /well HRP-conjugated anti-human IgG Fab antibody (Jackson ImmunoResearch Laboratories, catalogue #109-035-097) (5 $\mu\text{g}/\text{mL}$). The plates were washed with PBS six times and 100 μL /well ABTS solution (0.4 mg/mL 2,2'-azino-di(3-ethylbenzthiazoline-6-sulfonic acid), 0.003% H_2O_2 , 100 mM phosphate citrate, pH 4.0) was added for 30 min at room temperature. The absorbance of the wells at 405 nm was measured on a microplate reader. The initial and terminal half-lives of the PEG engagers were estimated by fitting the data to a two-phase exponential decay model with Prism 5 software (Graphpad Software).”

We added a description in the Supplementary Methods section (In vivo antitumor therapy).

“Groups of NOD SCID mice (n= 8) bearing $84.3 \pm 4.3 \text{ mm}^3$ subcutaneous MDA-MB-468 tumors on their right flank were intravenously injected with PBS, PEG engager^{EGFR} alone (6 mg/kg), free doxorubicin (3 mg/kg), 3 mg/kg Doxisome alone, PEG engager^{CD19} decorated Doxisome (1 mg/kg) or PEG engager^{EGFR} decorated Doxisome (1 mg/kg). Treatment was repeated once a week for a total of 4 weeks. Tumor sizes were measured every 7 days. Tumor volumes were calculated according the formula: length \times width \times height \times 0.5.“

We added a description in the Discussion section on page 20 line 369.

“We also demonstrated that pre-decoration of Doxil with PEG engager^{EGFR} could enhance the half-life of the engagers and improve the therapeutic index of Doxil treatment against EGFR⁺ human tumor xenografts, indicating that both pre-mixing and pretargeting of PEG engagers are promising therapeutic approaches.“

2) The work described in the manuscript is more a drug delivery method than a treatment modality for triple negative breast cancer. It is suggested to validate the platform in additional receptor/cancer systems, such as IGF1R and HER2, to demonstrate its general utility.

We appreciate this suggestion. IGF1R and HER may be good targets for other types of cancer. We plan to construct and test a HER2 targeted PEG engager in the future to validate whether this platform can be used for other targets such as HER2.

3) The efficacy differential between doxisome and doxisome/PEG engager treatments was only 2-fold in both MDA-MB-468 and MDA-MB-231 tumor models. Suggest carrying out a dose titration experiment for doxisome in the in vitro and in vitro models. The titration experiments will help to answer the question whether the PEG engager can reduce the concentration of doxisome in the treatment regimens.

We performed dose titration experiments as shown in the modified Figure 7. We found that a higher dose of Doxisome (3 mg/kg) is lethal to mice after the second administration as compared to a lower dose (1 mg/kg) (Fig. 7b). A higher dose of PEG engager (18 mg/kg) showed similar anti-tumor effects as compared to a lower dose of PEG engager (6 mg/kg) (Fig. 7a). The results of these studies are described in the Results section starting on page 12, line 223.

4) A naked doxorubicin control needs to be included in Figure 5a-c.

We included naked doxorubicin controls for anti-proliferation assays shown in Figure 5. The results show that free doxorubicin alone can efficiently inhibit cell proliferation as compared to treatment with Doxisome alone (Fig. 5a, 5b and 5c).

We added related description in the Results section (PEG engager-directed liposomal anti-cancer drugs can effectively inhibit proliferation of EGFR-positive cancer cells) on page 9 line 161

“... treated with graded concentrations of free drug (doxorubicin or vinorelbine), ...”

5) A PEG engager control was missing in Figure 7a (MDA-MB-468).

We included PEG engager alone in anti-tumor experiments. Tumor-bearing mice treated with PEG engager alone did not suppress tumor growth as compared to the treatment of PBS (Fig. 7a).

We modified the description in the Material and methods section on page 35 line 640.

We modified Results (Anti-tumor activity of pre-targeted PEG engager^{EGFR} combined with Doxisome) to include this result on page 13 line 227.

6) A naked doxorubicin control needs to be included in Figure 7a-c.

We repeated the in vivo experiments including free doxorubicin treatment in xenograft studies in Figure 7. A high dose of free doxorubicin (3 mg/kg) slightly inhibited the tumor growth as compared to PBS (green triangle in Fig. 7a). However, a lower dose of liposomal doxorubicin alone (1 mg/kg) showed better anti-tumor effects than free doxorubicin treatment (red square in Fig. 7a).

We modified the description in Materials and methods (In vivo antitumor therapy) on page 35.

A description of this experiment was added to the Results section (Anti-tumor activity of pre-targeted PEG engager^{EGFR} combined with Doxisome) on page 13 line 229.

Reviewer #4 (Remarks to the Author):

Roffler and co-workers have demonstrated the development and efficacy of a recombinant PEG engager^{EGFR} to specifically target PEGylated nanotherapeutics to EGFR-overexpressing TNBC cells. The authors convincingly show that the PEG engager^{EGFR} has high binding affinity to both PEG and EGFR, and that it efficiently binds to the TNBC cell surface without internalization until PEG is recruited and binds to the engager, with a preference for the high-density EGFR TNBC cells. For both in vitro and in vivo studies, the authors show an improvement in the anti-proliferative activity of PEGylated liposomal doxorubicin (Doxisome) when pre-targeted with PEG engager^{EGFR} compared to the Doxisome alone. The improved efficacy observed for NSG mice bearing MDA-MB-468 or MDA-MB-231 tumors (as

demonstrated by a reduction in tumor volume) was accompanied by limited side effects (indicated by no change in body weight). Appropriate negative controls were used throughout the study to show that the binding and efficacy of the PEG engagerEGFR was specific to the EGFR on TNBC cell surfaces.

The novelty of this work will be of great interest to pharmacologists, drug developers and clinicians alike. The significance of this work to the field includes the ability to implement existing, FDA-approved PEGylated therapeutics while potentially increasing efficacy at lower doses, which subsequently reduces side effects (e.g., HFS) and overall cost of treatment. Many investigators are developing approaches that functionalize the therapeutic in order to target the cancer cell, but are confronted with problems such as complexity of development and analytics, controlling the location and number of payload, specificity to the target, and reduction in 'stealth-ness'. The work presented by Roffler and colleagues largely bypasses these issues. Furthermore, a single PEGylated therapeutic can be utilized for heterogeneous tumors by incorporating engagers with specificities to other over-expressed proteins (other than EGFR) on cancer cell surfaces. Internalization of the engager-PEG complex correlates with the density of the overexpressed protein, thereby targeting the therapy to the cancer cells while potentially reducing effects on healthy cells. For these reasons, this paper will be a significant contribution to the field of anti-cancer drug development.

The methodology used to perform this study was appropriate, thorough, and utilized the appropriate techniques and controls. The data were presented in an order that created a natural flow. The data support the affinity of the engagers for their ligands, specificity to cell surface receptors, internalization correlating with the presence of PEG, enhancement of anti-tumor activity, tumor binding in vivo, and that efficacy is not compromised by pre-existing anti-PEG antibodies. The EGFR+ TNBC cell line MDA-MB-468 was used for most studies while other TNBC cell lines were used to study anti-proliferative activity of PEG therapeutics (MDA-MB-231 and BT20) and tumor size in in vivo studies (MDA-MB-231). Significant differences among data were evaluated appropriately using Student's t-test or by ANOVA with Dunnett's test. The conclusions drawn from the data are valid and well-supported by the data generated in the study. Proper credit has been given to previous work (to the best of my knowledge, along with my own literature searches).

The following includes a few questions/ comments that should be addressed by the authors to improve the paper. While I believe that the manuscript should be accepted without performing additional experiments, I have also provided suggestions for experiments to further characterize the efficacy and disposition of the

PEG-engagers.

1) How was the 150 ug dose of engager chosen? Why was the Doxisome normalized to mouse weight but the engagers dosed at a uniform amount to all mice? Has a maximum-tolerated dose been established for eth engagers? If so, what limits the maximum dose?

We selected what we thought was a reasonable dose of engager based on our previous experience using antibodies for in vivo therapy in mouse models. We performed an additional PEG engager dose titration experiments (Figure 7). A higher dose of PEG engager (18 mg/kg) showed similar anti-tumor effects as compared to a low dose of PEG engager (6 mg/kg). Therefore, we chose 6 mg/kg dose of PEG engager for additional experiments.

2) The recombinant PEG engagers appear larger than the expected MW of 85 kDa. Why is this?

The protein ladder was run under reducing conditions and may not be very accurate for proteins run under non-reducing conditions. Therefore, we included Eribtux IgG as a positive protein control (~150 kDa). Fig. 2 shows that Eribtux appears larger than the expected MW of 150 kDa (~200 kDa), illustrating the limitations of using protein markers to estimate protein sizes under non-reducing conditions.

We further determined the precise molecular weight of PEG engagers by MALDI-TOF. The results show that the molecular weights of PEG engager^{EGFR} and PEG engager^{CD19} are 79 kDa and 78 kDa, respectively (Fig. 2c).

We modified the description in the Results section (Characterization of bispecific PEG engagers) on page 6 line 106 and added a description in Materials and Methods (Analysis of purified PEG engager antibodies by SDS-PAGE and MALDI-TOF) on page 27 line492.

3) Would the Alexa Fluor label affect binding of the engagers to their ligands?

To address this issue, we added serial dilutions of PEG engagers or Alexa Fluor-labeled PEG engagers (A647 PEG engagers) to 4arm-PEG_{10k} immobilized in EIA plates followed by detection of engager binding with HRP conjugated goat

anti-human Fab antibodies. As the result show below, A647-labeled PEG engagers showed similar PEG-binding activity as compared to parental PEG engagers.

4) The PK results for both PEG engager^{EGFR} and PEG engager^{CD19} show that approximately 10% of the engager is present in plasma at 5 hours and detectable levels are observed in the plasma at 24 hours. If the PEGylated liposomal doxorubicin is injected at 5 hours, it is likely that the engager + Doxisome will bind and circulate in the blood. Are there any toxic effects expected for this circulating engager + NP? How is the PK of the engager and/or Doxisome affected by this interaction in the blood?

We don't expect toxicity by circulating engagers because they are monovalent and should not induce aggregation of the nanoparticles. To further address this issue, we pre-mixed PEG engagers with Doxisome using a ratio of PEG engager and PEG-lipid on Doxisome of 1:55 (Supplementary Fig. 11a). The half-lives of the PEG engagers were increased to approximately 3.5 h (PEG engager^{EGFR}) and 3.8 h (PEG engager^{CD19}) (Supplementary Fig. 11b) after i.v. administration of PEG engager-docked Doxisomes (containing 30 µg PEG engagers).

We also performed an anti-tumor activity experiments using the pre-mix strategy and we didn't observed toxic effects as the body weight change of mice were stable after treatments(Supplementary Fig. 11d). These results are consistent with lack of toxicity by circulating PEG engagers.

5) Several figures refer to an n=3, but error bars are not included in the plots (or they are small enough so that I cannot see them). Please check error bars on the following figures

- Figure 2d (CD19)

- Figures 5a-c
- Figure 6a
- Supplementary figure 2a

We thank reviewer to point out these problems. The errors bars are not visible if we use SEM. We presented the results as the mean \pm SD.

6) The authors state in lines 249-251 that '...antibodies that recognize the extracellular domain of the EGFR can target both wild-type and mutated EGFRs.' Has the binding of engager, and, subsequently, the PEG therapeutic, been demonstrated for models that bear activating EGFR mutations in TNBC?

To address this issue, we examined the anti-proliferation activity of PEG-liposomal doxorubicin in PC9 cells (EGFR⁺, lung cancer cells with mutant EGFR, exon19del E746–A750). The results show that PEG engager^{EGFR} specifically targeted EGFR-positive PC9 cells and increased the anti-proliferation activity of Doxisome (Supplementary Fig. 4b).

We added the related description in Materials and method (Cell proliferation assay) on page 32 line 585

We also modified the description in the Results section (PEG engager-directed liposomal anti-cancer drugs can effectively inhibit proliferation of EGFR-positive cancer cells) on page 9 line 156.

7) It would be very interesting to evaluate the disposition of the engagers in tissue - has this been evaluated? Furthermore, it is well understood that the MPS system is involved in the clearance of nanoparticles and biotherapeutics. Have the authors evaluated the interaction of the engagers with the MPS? This would be valuable information in understanding the PK of the engagers.

This is an excellent suggestion. In the future, we would like to radio-label or fluorescence-label the PEG engagers for tracking their bio-distribution in vivo and evaluating the interaction of the PEG engagers with the MPS.

8) Would the authors comment on the presence of 4armPEG10k-NIR-797 in the MDA-MB-468 xenograft tumors in NSG mice that were dosed with PEG engager^{CD19} (negative control; see Figure 6b). Is this attributed to EPR of the PEG

structure, or is it due to binding of the PEG engager^{CD19} to the tumor? Furthermore, why is there fluorescence observed in the head/neck region for 3 of 6 mice receiving PEG engager^{EGFR}?

We observed that PEG-NIR-797 can spontaneously accumulate in the tumors from our previous study (Mol. Cancer Ther., 9: 1903-1912, 2010). We think this might be attributed to EPR effect of the PEG size and structure. We repeated the in vivo imaging studies but did not observe specific uptake of fluorescence in the head and neck regions of the mice.

9) Treatment of TNBC brain metastases is a topic of current research for PEGylated liposomal doxorubicin. Have the authors performed studies in intracranial models? Is there any evidence of how the engagers affect drug accumulation within the blood brain barrier?

In fact, our preliminary results (unpublished data) suggest that PEGylated compounds (e.g., PEG-protein, PEG-nanoparticle and PEG-fluorescent dye) can spontaneously accumulate in the brain as compared to the parental compounds. Thus, we hypothesized that PEG might be very helpful for compound delivery of BBB. It is unclear if this is a specific effect or is due to increased circulation times afforded by PEGylation. We also performed pilot experiments to see if the engagers increased BBB passage but could not find evidence for this.

We thank the reviewer for their careful reading and detailed suggestions and corrections to improve the manuscript. ,

Figure 2. Production and analysis of recombinant PEG engagers. (a) Schematic representation of PEG engager constructs, which code for a murine Ig kappa chain leader sequence (Igκ), a humanized 6.3 light chain (6.3VL-Cκ), an IRES sequence, a humanized 6.3 heavy chain (6.3VH-CH₁), a glycine-serine peptide linker (GGGGS), an anti-tumor disulfide-stabilized scFv (anti-CD19 or anti-EGFR), and a polyhistidine-tag (His tag). (b) Reducing (left panel) and non-reducing (right panel) SDS PAGE showing Coomassie Blue staining of Erbitux, PEG engager^{EGFR} and PEG engager^{CD19}. M, PageRuler prestained protein ladder (Fermentas). (c) Precise molecular weight of PEG engager^{EGFR} (left panel) and PEG engager^{CD19} (right panel) analyzed by MALDI-TOF. (d) Binding affinity of PEG engager^{EGFR} (left panel) and PEG engager^{CD19} (right panel) to PEG_{5k} analyzed by microscale thermophoresis (n= 3). (e) Binding affinity of PEG engager^{EGFR} to recombinant EGFR protein (left panel) and PEG engager^{CD19} to recombinant CD19 protein (right panel) analyzed by microscale thermophoresis (n= 3). Data are shown as mean ± SD. Representative microscale thermophoresis data from three independent experiments are shown.

Figure 3. Dual antigen-binding activity of PEG engagers. PEG engager^{EGFR} (upper panels) and PEG engager^{CD19} (lower panels) supplemented with Hoechst 33342 (blue) were incubated with MDA-MB-468 (a), A431 (b) or MCF7 (c) cells followed by PEG-Qdot655 (red) and observed in real time with a digital confocal microscope. Scale bars, 10 μm. Representative confocal images from three independent experiments are shown.

Figure 4. Conditional internalization of PEGylated nanoparticles by pre-targeted PEG engager^{EGFR}. Pre-targeted fluorescent-labeled PEG engager^{EGFR} (green) on the cell membrane of MDA-MB-468 cells at 37°C for 1 h was real-time imaged after incubating with (b) or without (a) PEG-QD655 (red). Cells were further imaged 5 min, 30 min and 60 min after addition of medium alone or PEG-QD655 (red). Hoechst 33342 (blue) and LysoTracker Red DND-99 (appears in purple pseudo color) for nucleus and lysosome staining, respectively. Scale bars, 10 μ m. (c) Percentage of the PEG engager^{EGFR} that internalized into cells complexed with (□) or without (■) PEG-QD655 at different times was quantified from confocal images of individual cells (n=15). (d) Percentage of PEG engager^{EGFR} that co-localized to lysosomes at different times was quantified from confocal images of individual cells (n=15). Representative confocal images from two independent experiments are shown. Data are shown as mean \pm SD. Significant differences in percentage of PEG engager^{EGFR} internalization or lysosome co-localization with and without addition of PEG-QD655 are indicated: ***, $p \leq 0.0005$ (Student's t-test). N.S., not significant.

Figure 5. PEG engager^{EGFR} enhances the anti-proliferative activity of drug loaded PEG-liposomes. BT-20 (a), MDA-MB-468 (b), and MDA-MB-231 cells (c) were incubated with PEG engager^{EGFR} (○), PEG engager^{CD19} (■), or culture medium (●) for 30 min followed by serial dilutions of free doxorubicin (△), Doxosome (liposomal doxorubicin), PEG engager^{EGFR} alone (◇) or empty liposomes (□) in triplicate for 4 h. The incorporation of ³H-thymidine into cellular DNA was measured 72 h later. The data are representative of three independent experiments. (d) The EC₅₀ values of BT-20, MDA-MB-468 and MDA-MB-231 cells treated with PEG engager^{CD19} plus Doxosome or PEG engager^{EGFR} plus Doxosome were analyzed (n = 3). Data are shown as mean ± SD. Significant differences in mean EC₅₀ values are indicated: ***, p ≤ 0.0005 (Student's t-test).

Figure 6. Pharmacokinetics and imaging of PEG engagers in mice. (a) NSG mice were i.v. injected with 6 mg/kg PEG engager^{EGFR} or PEG engager^{CD19}. Mean plasma concentrations of the PEG engagers were measured by sandwich ELISA (n = 3 mice). (b) Five hours before i.v. administration of 4armPEG_{10k}-NIR-797 probes (5 mg/kg), NSG mice bearing s.c. MDA-MB-468 tumors were i.v. injected with 6 mg/kg PEG engager^{EGFR} or PEG engager^{CD19} and the whole-body imaging were sequentially imaged at 24, 48 and 72 h with an IVIS spectrum optical imaging system. (c) The uptake of PEG-NIR797 in MDA-MB-468 tumors was determined by measuring fluorescence intensities (n = 3). Data are shown as mean ± SD. Significant differences in mean fluorescent intensity between PEG engager^{EGFR} and PEG engager^{CD19} groups are indicated: *, p ≤ 0.05 (Student's t-test).

Figure 7. Therapeutic efficiency of PEG engager directed Doxisome. Five hours before administration of 1 mg/kg/dose or 3 mg/kg/dose Doxisome (◆), groups of six or eight SCID mice bearing MDA-MB-468 (a) or MDA-MB-231 tumors (c) were i.v. injected with free doxorubicin (▲), 6 mg/kg PEG engager^{EGFR} (○), 18 mg/kg PEG engager^{EGFR} (□) or 6 mg/kg PEG engager^{CD19} (■) once a week for 4 weeks (arrows). For negative controls, groups of untreated mice received PBS (●) or PEG engager^{EGFR} alone (◇). Results show mean tumor sizes (n = 8 for MDA-MB-468, n = 6 for MDA-MB-231 (43 days post first treatment)). Data are shown as mean ± SD. (b) Mean body weights of treated MDA-MB-468 mice (n = 8). Statistical analysis of the differences in tumor volumes between treatment and control groups was performed by one-way analysis of variance (ANOVA) followed by Dunnett's multiple comparisons. *, p ≤ 0.05, **, p ≤ 0.005.

Supplementary Figure 2. Conditional internalization of PEGylated nanoparticles by pre-targeted PEG engager^{EGFR} in EGFR-positive cells. Pre-targeted fluorescent-labeled PEG engager^{EGFR} (green) on the cell membrane of BT20 cells was real-time imaged after incubating without (a) or with (b) PEG-QD655 (red). The cells were further imaged at 5 min, 30 min and 60 min after addition of medium alone or PEG-QD655 (red). Hoechst 33342 (blue) and LysoTracker Red DND-99 (appears in purple pseudo color) for nucleus and lysosome staining, respectively. Scale bars, 10 μ m.

Supplementary Figure 3. Conditional internalization of PEGylated nanoparticles by pre-targeted PEG engager^{EGFR}. Pre-targeted fluorescent-labeled PEG engager^{EGFR} (green) on the cell membrane of MDA-MB-468 cells was real-time imaged 1 h (a) or 9 h (b) after antibody addition. Cells were further imaged 1 min, 30 min and 60 min after addition of PEG-QD655 (red). Hoechst 33342 (blue) for nucleus staining. Scale bars, 10 μ m. (c) Percentage of the PEG engager^{EGFR} (left panel) or PEG-QD655 (right panel) that internalized into cells at different times was quantified from confocal images of individual cells (n=16). Representative confocal images from two independent experiments are shown. Data are shown as mean \pm SD. Significant differences in percentage of internalization before and after addition of PEG-QD655 are indicated: ***, $p \leq 0.0005$ (Student's t-test).

Supplementary Figure 4. PEG engager^{EGFR} enhances the anti-proliferative activity of PEG-liposomal doxorubicin in EGFR-positive cancer cells. SKBR3 (a), PC9 (b), and HepG2 cells (c) were incubated with PEG engager^{EGFR} (○), PEG engager^{CD19} (■), or culture medium (●) for 30 min followed by serial dilutions of **free doxorubicin** (△), Doxisome (liposomal doxorubicin) or empty liposomes (□) in triplicate for 4 h. The incorporation of ³H-thymidine into cellular DNA was measured 72 h later. The data are representative of three independent experiments. (d) The EC₅₀ values of SKBR3, PC9 and HepG2 cells treated with PEG engager^{CD19} plus Doxisome or PEG engager^{EGFR} plus Doxisome were analyzed (n = 3). Data are shown as mean ± SD. Significant differences in mean EC₅₀ values are indicated: ***, p ≤ 0.0005 (Student's t-test). N.S., not significant.

Supplementary Figure 5. PEG engager^{EGFR} enhances the anti-proliferative activity of vinorelbine loaded PEG-liposomes. BT-20, MDA-MB-468, and MDA-MB-231 cells were incubated with PEG engager^{EGFR} (□) or PEG engager^{CD19} (■) for 30 min followed by serial dilutions of Lipo-Vino (liposomal vinorelbine) in triplicate for 4 h. The incorporation of ³H-thymidine into cellular DNA was measured 72 h later. The data are representative of three independent experiments. The EC₅₀ values of BT-20, MDA-MB-468, and MDA-MB-231 cells treated with PEG engager^{CD19} plus Lipo-Vino or PEG engager^{EGFR} plus Lipo-Vino were analyzed (n = 3). Data are shown as mean ± SD. Significant differences in mean EC₅₀ values are indicated: *, p ≤ 0.05; **, p ≤ 0.005 (Student's t-test).

Supplementary Figure 6. Down-regulation of EGFR decreases PEG engager^{EGFR} mediated anti-proliferative activity of PEG-liposomal doxorubicin. BT20/shEGFR cells were incubated with PEG engager^{EGFR} (○), PEG engager^{CD19} (■), or culture medium (●) for 30 min followed by serial dilutions of free doxorubicin (△), Doxosome (liposomal doxorubicin) or empty liposomes (□) in triplicate for 4 h. The incorporation of ³H-thymidine into cellular DNA was measured 72 h later. The data are representative of three independent experiments. Data are shown as mean ± SD.

Supplementary Figure 8. Imaging of PEG engagers in A431 tumor-bearing mice.

(a) Five hours before i.v. administration of 4armPEG_{10k}-NIR-797 probes (5 mg/kg), BALB/c nude mice bearing s.c. A431 tumors were i.v. injected with 6 mg/kg PEG engager^{EGFR} or PEG engager^{CD19} and the whole-body imaging were sequentially imaged at, 24, 48 and 72 h with an IVIS spectrum optical imaging system. (b) The uptake of PEG-NIR797 in A431 tumors was determined by measuring fluorescence intensities (n = 3). Data are shown as mean ± SD. Significant differences in mean fluorescent intensity between PEG engager^{EGFR} and PEG engager^{CD19} groups are indicated: *, p ≤ 0.05 (Student's t-test).

Supplementary Figure 9. Imaging of PEG engagers in HepG2 tumor-bearing mice. (a) Five hours before i.v. administration of 4armPEG_{10k}-NIR-797 probes (5 mg/kg), BALB/c nude mice bearing s.c. HepG2 tumors were i.v. injected with 6 mg/kg PEG engager^{EGFR} or PEG engager^{CD19} and the whole-body imaging were sequentially imaged at 24, 48 and 72 h with an IVIS spectrum optical imaging system. (b) The uptake of PEG-NIR797 in HepG2 tumors was determined by measuring fluorescence intensities (n = 3). Data are shown as mean ± SD. Significant differences in mean fluorescent intensity between PEG engager^{EGFR} and PEG engager^{CD19} groups are indicated: N.S., not significant (Student's t-test).

Supplementary Figure 10. PEG engager^{EGFR} blocks the EGFR signaling pathway in EGFR-positive cells. Starved A431 cells (24 hours) were incubated with or without EGF and then sequentially treated with PEG engager^{CD19}, PEG engager^{EGFR}, Herceptin or Erbix. Phosphorylation of EGFR and Erk were detected by western blotting using anti-phospho EGFR or anti-phospho ERK antibodies. Total EGFR and tubulin served as loading controls.

Supplementary Figure 11. Therapeutic efficiency of PEG engager modified Doxosome. (a) PEG engager^{EGFR} and Doxosome was premixed in different molar ratios of PEG engager and PEG-lipid on Doxosome ranging from 1: 18.3 to 1: 110 at 4 °C for 1 hour. MDA-MB-468 cells were incubated with serial dilutions of Doxosome alone (∇), pretargeted engager^{EGFR} followed by Doxosome (\diamond) or premixed PEG engager^{EGFR} / PEG-lipid on Doxosome (1:18.3) (\bullet), (1:27.5) (\blacksquare), (1:37.6) (\circ), (1:55) (\square), (1:110) (\triangle) in triplicate for 4 h. The incorporation of ³H-thymidine into cellular DNA was measured 72 h later. (b) NOD SCID mice were i.v. injected with PEG engager decorated Doxosome (3 mg/kg). Mean plasma concentrations of the PEG engagers were measured by sandwich ELISA (n = 3 mice). (c) PEG engagers were pre-mixed with Doxosome at a molar ratio 1:55. Groups of eight NOD SCID mice bearing MDA-MB-468 tumors were i.v. injected with saline (\bullet), 6 mg/kg PEG engager^{EGFR} (\diamond), 1 mg/kg PEG engager^{EGFR}/Doxosome (\circ), or 1 mg/kg PEG engager^{CD19}/Doxosome (\blacksquare), 3 mg/kg PEG engager^{EGFR}/Doxosome (\blacklozenge) or 3 mg/kg PEG engager^{CD19}/Doxosome (\square) once a week for 4 weeks (arrows). Results show mean tumor sizes (n = 8). Data are shown as mean \pm SD. (d) Mean body weights of treated MDA-MB-468 mice (n = 8). Statistical analysis of the differences in tumor volumes between treatment and control groups was performed by one-way analysis of variance (ANOVA) followed by Dunnett's multiple comparisons. *, p \leq 0.05, **, p \leq 0.005.

REVIEWERS' COMMENTS:

Reviewer #2 (Remarks to the Author):

Although the revised manuscript demonstrates function in vivo in one model not site-relevant for breast cancer, the authors have revised their manuscript to address all other comments of this reviewer.

Reviewer #3 (Remarks to the Author):

The authors have addressed all my questions in the revised manuscript.

Reviewer #4 (Remarks to the Author):

I am very satisfied with the responses that the authors provided to their reviewers. Each point was addressed and thoughtfully answered. Additional experiments were conducted in some cases to provide evidence for their conclusions. In my opinion, the manuscript is acceptable for publication.